# High-throughput mechanical phenotyping and transcriptomics of single cells

Akifumi Shiomi [1,2], Taikopaul Kaneko [1], Kaori Nishikawa[1], Arata Tsuchida [1], Takashi Isoshima [1], Mayuko Sato [3], Kiminori Toyooka [3], Kentaro Doi [4], Hidekazu Nishikii[5] & Hirofumi Shintaku [1,2] ✉

The molecular system regulating cellular mechanical properties remains unexplored at single-cell resolution mainly due to a limited ability to combine mechanophenotyping with unbiased transcriptional screening. Here, we describe an electroporation-based lipid-bilayer assay for cell surface tension and transcriptomics (ELASTomics), a method in which oligonucleotide-labelled macromolecules are imported into cells via nanopore electroporation to assess the mechanical state of the cell surface and are enumerated by sequencing. ELASTomics can be readily integrated with existing single-cell sequencing approaches and enables the joint study of cell surface mechanics and underlying transcriptional regulation at an unprecedented resolution. We validate ELASTomics via analysis of cancer cell lines from various malignancies and show that the method can accurately identify cell types and assess cell surface tension. ELASTomics enables exploration of the relationships between cell surface tension, surface proteins, and transcripts along cell lineages differentiating from the haematopoietic progenitor cells of mice. We study the surface mechanics of cellular senescence and demonstrate that *RRAD* regulates cell surface tension in senescent TIG-1 cells. ELASTomics provides a unique opportunity to profile the mechanical and molecular phenotypes of single cells and can dissect the interplay among these in a range of biological contexts.

Cellular mechanical properties are known to couple with functions in stem cells[1–4], cancer cells[5–7], and senescent cells[8–10]. Recent technical advances in mechanical phenotyping of single cells[11–14] have offered opportunities to profile mechanical hallmarks of cellular functions and states as information orthogonal to the characterization of cellular states by molecular abundance, that is, by molecular phenotype[15]. However, current techniques have a limited ability to interrogate the link between the mechanical and the molecular phenotype as defined by unbiased transcriptional screening[16,17]. Thus, the underlying molecular system remains largely unexplored at a single-cell resolution[18]. Here, we describe ELASTomics, a method that can integrate phenotyping of cell surface mechanics with unbiased transcriptomics for thousands of single cells. ELASTomics utilizes nanopore electroporation, which imports molecules into cells in a manner dependent on the cell surface tension[19–23]. We demonstrate that this method is readily adaptable to cultured cell lines and primary cells and show that joint analysis uncovers the transcriptional regulation of cell surface mechanics in various biological contexts, including cancer malignancy, differentiation of haematopoietic cells, and cellular senescence.

## Results

### Strategy of ELASTomics

Our strategy for characterising the mechanical state of the cell surface involved permeabilizing plasma membranes of living cells using

[1]Cluster for Pioneering Research, RIKEN, Saitama, Japan. [2]Institute for Life and Medical Sciences, Kyoto University, Kyoto, Japan. [3]Center for Sustainable Resource Science, RIKEN, Yokohama, Japan. [4]Department of Mechanical Engineering, Toyohashi University of Technology, Toyohashi, Japan. [5]Faculty of Medicine, University of Tsukuba, Tsukuba, Japan. ✉e-mail: shintaku@infront.kyoto-u.ac.jp

nanopore electroporation[21,24], electrophoretically importing DNA-tagged dextran (DTD) molecules with various Stokes radii ($4.1 \pm 0.0$ nm–$17.0 \pm 12.2$ nm, Supplementary Table 1), and digitally counting imported DTD molecules using sequencing (Fig. 1a and Supplementary Fig. 1). As the size of the pore formed in a plasma membrane under nanopore electroporation has been shown to increase with plasma membrane tension[24,25], we hypothesized that the number of imported DTD molecules reflects cell surface mechanics. To enable simultaneous detection of DTD and mRNA using existing single-cell sequencing methods, we designed oligonucleotides for DTD that can be captured by oligo-polythymidine primers and contain a barcode

sequence for DTD identification and a handle for amplification using polymerase chain reaction (PCR) (Supplementary Table 2). In ELAS-Tomics, we seeded cells on a track-etched membrane that embedded multiple nanopores 100 nm in diameter (Supplementary Fig. 2a-d). We placed a buffer solution containing DTD on the non-seeded side of the membrane (Supplementary Fig. 2e-g, Supplementary Table 3) and then applied pulsed voltages (40 V, 5 ms width, square waves at 20 Hz, 500 cycles, unless otherwise specified) across the track-etched membrane with a pair of electrodes (Supplementary Fig. 2h, i) to generate a focused electric field in the vicinity of the nanopores (Supplementary Fig. 2j-l). The focused electric field reversibly electroporated the

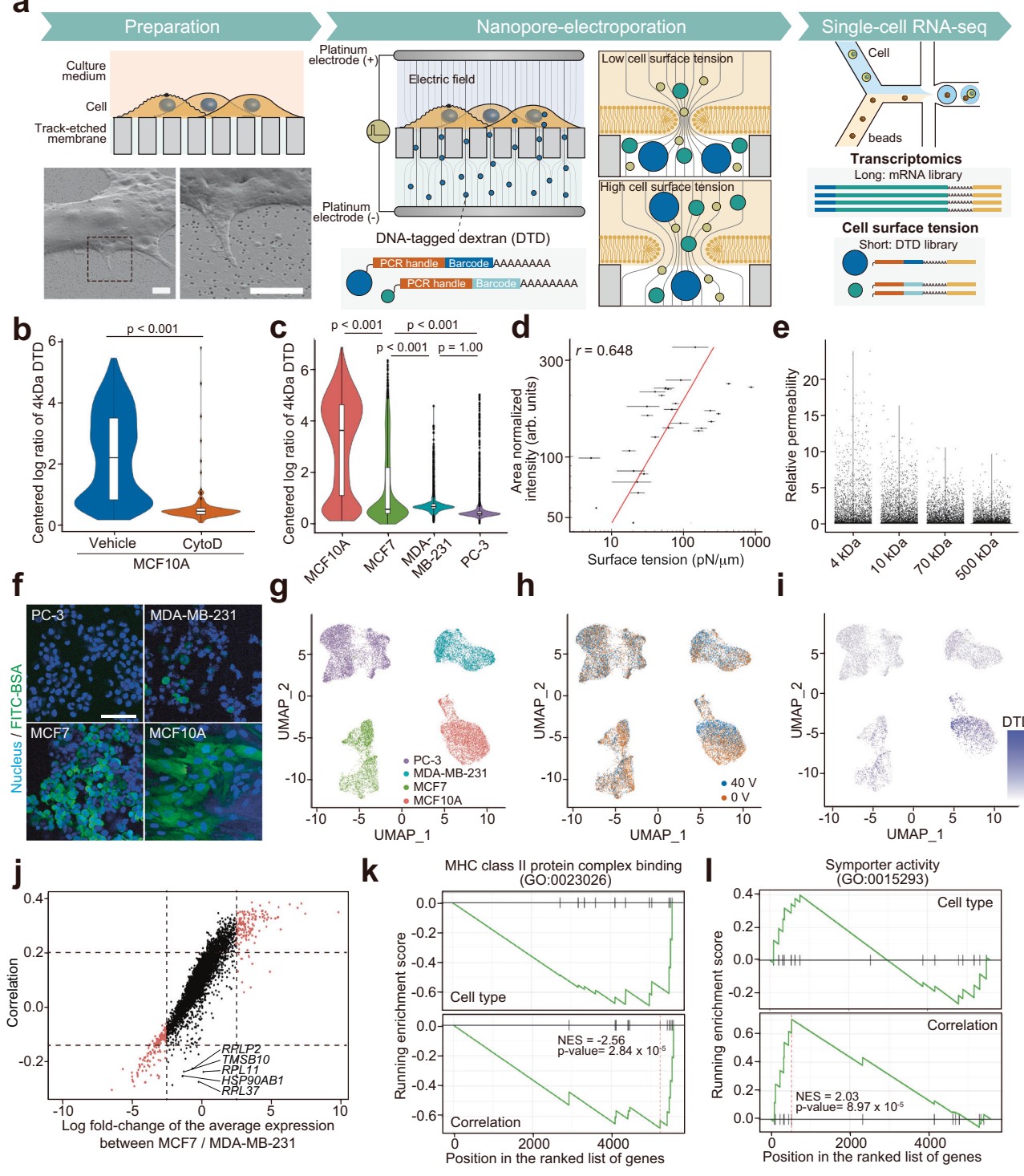

**Fig. 1 | ELASTomics enables simultaneous detection of single-cell surface tension coupled with transcriptomes. a** Workflow of ELASTomics. (*Preparation*) Cells are cultured on the track-etched membrane coated by fibronectin. The lower left panel is field emission scanning electron microscopy (FE-SEM) images of MCF7 cells on a track-etched membrane, and the lower right panel is zoom-up views of the dotted box in the lower left panel. An independent duplicate experiment has verified similar results at a different angle of FE-SEM imaging. Scale bars, 2 μm. (*Nanopore-electroporation*) Cells are subjected to pulsed electric fields that are created in the vicinity of nanopores to import DNA-tagged dextran (DTD) into the cells. The cell surface tension of the lipid bilayer affects the diameter of the pore formed in a lipid bilayer. The amount of imported DTD thus dictates the surface tension of individual cells. (*Single-cell RNA-seq*) Single-cell RNA-sequencing (scRNA-seq) workflow creates libraries of transcriptomics (mRNA library) and cell surface tension (DTD library). **b** Centered log ratios of 4 kDa DTD imported into MCF10A cells with or without cytochalasin D treatment. **c** Centered log ratios of 4 kDa DTD imported into four cell lines (MCF10A, MCF7, MDA-MB-231, and PC-3). For boxplot overlaid on a violin plot (b and c), the center line is the median, the box indicates the first and third quartiles, whiskers are minimum/maximum values excluding outliers, and dots are outliers. **d** Normalized intensity of FITC-labeled bovine serum albumin (FITC-BSA) and cell surface tension measured by atomic force microscopy in each nanopore-electroporated MCF10A cell ($n = 28$ independent cells examined

in four independent experiments). The coefficient of Pearson's correlation (two-tailed) is $r = 0.648$. Error bars of surface tension are presented as mean ± SE for $n = 2$ to 44 times of each cell. The red line represents the regression curve. **e** Relative permeability of different DTDs (4, 10, 70, and 500 kDa) imported into MCF10A cells. Counts are respectively normalized by concentration and the mobility of each DTD. **f** Fluorescence images of nanopore-electroporated cells with FITC-BSA (green). Cell nuclei are stained by Hoechst 33342 (blue). An independent duplicate experiment has verified the similar heterogeneity across cell types in FITC-BSA measurements by flow cytometry. Scale bar, 10 μm. **g–i** Uniform manifold approximation and projection (UMAP) of cells. The color indicates the cell types (purple: PC-3; blue: MDA-MB-231; green: MCF7; red: MCF10A) (**g**) and the applied voltages (red: 0 V; blue: 40 V) of nanopore-electroporation (**h**), and centered log ratios of 4 kDa DTD imported into cells by nanopore-electroporation (**i**). **j** Coefficient of correlation between centered log ratio of DTD and the expression of individual genes (Correlation) plotted against the log fold-change of the gene expression between MCF7 and MDA-MB-231 (Cell type). Red points are genes with the log fold-change in average expression is >2.5 or <−2.5. **k, l** Gene set enrichment analysis (GSEA) showing enriched pathways with genes ordered by correlation and cell type shown in **j**. The P values (*p*) are indicated in the graph (**b**: two-tailed Student's t-test; **c**: Tukey's t-test). Source data are provided as a Source Data file.

plasma membrane in the proximity of the nanopores and imported DTD by electrophoresis. Unlike in bulk electroporation, the electric field focused at the nanopores offers efficient[21,24] and robust electroporation irrespective of cell size, shape, or orientation[26,27] and electroporates the plasma membrane in a surface-tension-dependent manner. In the present study, we demonstrate the integration of ELASTomics with single-cell RNA-sequencing (scRNA-seq) using 10x Genomics Single Cell 3′ v3.1 to simultaneously profile the mechanical state and transcriptomic expression in thousands of single cells, leveraging the CITE-seq (cellular indexing of transcriptomes and epitopes with sequencing) protocol[28] to amplify DNA products derived from cDNA and DTD and construct two Illumina sequencing libraries.

## Optimization of nanopore electroporation for ELASTomics
To balance between the sensitivity and the negative impact of nanopore electroporation on the cells, we assessed the quantity of imported molecules and cellular viability by performing the nanopore electroporation with fluorescein isothiocyanate-labelled bovine serum albumin (FITC-BSA; Stokes radius of $3.6 ± 1.4$ nm) as a substitute of DTD at various voltage conditions followed by flow cytometry (Supplementary Fig. 3a, b). The amount of imported FITC-BSA correlated well with that of DTDs (Supplementary Fig. 4a-d). To prevent cell stimulation and cell death due to calcium influx, we selected calcium-free solutions, PBS (-) and HEPES-based buffer (20 mM HEPES/NaOH, pH 7.0 and 260 mM sucrose), for nanopore electroporation. Our experimental data showed that the higher applied voltage increased the amount of imported molecules, improving the sensitivity of the measurement of the cell surface tension (Supplementary Fig. 3c). However, the excessive magnitude of the applied voltage gave a negative impact on the cell viability and distorted the gene expression (Supplementary Fig. 3d-g). On the basis of the experimental surveillance, we determined the conditions of ELASTomics with >90% viability as 40 V for the four cancer cell lines (PC-3, MDA-MB-231, MCF7, and MCF10A cells), 75 V for mouse haematopoietic stem/progenitor cells (mHSPCs) and 50 V for TIG-1 cells (Supplementary Fig. 3h-m). The quantity of the imported molecules was reproducible under the same condition (Supplementary Figs. 3a, b, and 5e, f). We confirmed that nanopore electroporation can be applied to various cell lines including HeLa, PC-3, MDA-MB-231, MCF7, MCF10A, TIG-1, OVCAR-3 (adherent cells, human), CHO-K1 (adherent cells, Chinese hamster), GEM-81 cells (adherent cells, goldfish), primary mHPSCs (suspension cells, mouse), and K562 (suspension cells, human) (Supplementary Fig. 3h-r). We note that non-adherent cells such as K562 and mHSPCs require higher

applied voltages than adherent cells to import a similar quantity of molecules by nanopore electroporation. We also note that incubation for 2 minutes after applying the electrical pulses was critical to reseal the pores in the phospholipid bilayers and to protect the cells from calcium stimulation and cell death. After trypsinization, cells were kept on ice to suppress changes in gene expression and performed scRNA-seq within 1 h. To further minimize the effect of the nanopore-electroporation and the DTD import on the gene expression analysis, we collected single-cell data with non-nanopore electroporated cells and integrated with the ELASTomics data before analysis (Fig. 1b,c and Supplementary Fig. 3s-u). This process allowed us to robustly analyse the gene expression.

## Heterogeneity of cell surface tension
According to a theoretical understanding of the electroporation process (Supplementary information)[25,29], surface tension influences the size of a pore in a plasma membrane via free energy changes during the formation of a pore by the release of surface energy (Supplementary Fig. 5a). At the same transmembrane voltage and line tension, higher surface tension increases the probability of forming a larger pore (Supplementary Fig. 5b). The number of DTD molecules imported into cells thus increases with cell surface tension (Supplementary Fig. 5c, d). To test whether the cell-to-cell variability reflected by DTD abundance corresponds to quantitative differences in the cell surface tension of individual cells, we investigated the relationship between the number of molecules imported and the surface tension of individual MCF10A cells by measuring these parameters using fluorescence microscopy and atomic force microscopy (AFM)[30] (Supplementary Fig. 4e-g), respectively. Although the measurements of surface tension using AFM were performed between 0.5–3 h after nanopore electroporation, we observed a correlation between the area-normalized quantity of imported FITC-BSA and the surface tension measured by AFM (Fig. 1d, and Supplementary Fig. 4h, i), indicating that the variation in the number of imported molecules reflected cell-to-cell variation in surface tension.

Interestingly, the total quantity of imported molecules into a cell by nanopore electroporation showed a comparable correlation with the surface tension (Supplementary Fig. 4h), although it is predicted to be proportional to the adhesion area, i.e., the number of nanopores in the track-etched membrane beneath the cell membrane. We attribute this to the dependence of the adhesion area on the cell surface tension[31], which was also observed in our data (Supplementary Fig. 4i), supporting that the effect of the adhesion area on nanopore-electroporation

reinforces, rather than weakens, the correlation between surface tension and the amount imported molecules.

## Effect of Stokes radius of DTD on its translocation

We next investigated the effect of the Stokes radius of DTD on electrophoretic translocation through a pore in a lipid bilayer. We designed the DTD so that the Stokes radii ($4.1 \pm 0.0$ nm–$17.0 \pm 12.2$ nm) cover the critical radius, 15 nm, which is the theoretical maximum radius to reseal at equilibrium[24]. Here, we defined the relative permeability of the DTD as counts/$\mu_{DTD} c_{DTD}$, where $\mu_{DTD}$ and $c_{DTD}$ are the electrophoretic mobility and the concentration of DTD in the buffer, respectively. The permeability of various sizes of DTD decreased as the Stokes radius increased (Fig. 1e), indicating that steric hindrance due to DTD size reduced translocation. Notably, the permeability of DTDs with different molecular weights was highly consistent (Supplementary Fig. 6), supporting the reliability of the DTD counts for the quantification regardless of the cell types. The permeability ratio for a DTD pair was independent of cell type (Supplementary Fig. 6), implying that the effect of cell size can be normalized by the permeability ratio. However, we found that the permeability ratio was susceptive to noise than DTD count with our dataset. We thus primary used normalized 4 kDa DTD counts using centred log ratio transformation to correlate the cell surface tension on the basis of the correlation between the total quantity and the surface tension (Supplementary Fig. 4h). The expected error in the measurement of the cell surface tension using the total quantity of the imported molecules was estimated on the order of $10^{\pm 0.39}$ fold in the range of 10-1000 pN/$\mu$m on the basis of the correlation among them.

## Cell surface tension–cortical tension and plasma membrane tension

To assess the utility of ELASTomics in profiling the surface mechanics of single cells, we designed two proof-of-principle experiments investigating cells with a range of cell surface tensions influenced by cortical and plasma membrane tension[7,32,33]. We perturbed the cell surface tension of MCF10A cells using cytochalasin D, which reduces the cell surface tension under control by cortical tension by inhibiting actin polymerization, disrupting filamentous actin structures. Following cytochalasin D treatment or vehicle on MCF10A cells, we performed ELASTomics with the respective MCF10A cells, where we prepared scRNA-seq libraries and counted the number of DNA tags derived from DTD in single cells (Fig. 1b). As expected, MCF10A cells treated with cytochalasin D showed lower DTD counts than did control cells treated with the vehicle, consistent with the theoretical prediction. Additionally, we confirmed that the nanopore-electroporation can probe the perturbation in the cell surface tension by blebbistatin and Y-27632, which changes the cortical tension, (Supplementary Fig. 5e), and methyl-β-cyclodextrin, which changes the plasma membrane tension (Supplementary Fig. 5f), suggesting the robustness of the approach.

In another proof-of-principle experiment, we applied ELASTomics to four cell lines with varying plasma membrane tensions that are coupled with cancer malignancy and invasion[7]. The mechanical properties of cancer cells have an important influence on metastatic dissemination[5] and cancer stemness[6]. A recent study demonstrated that low plasma membrane tension is a prerequisite for high invasiveness in cancer cells[7]. For this experiment, we used non-invasive epithelial cells (MCF10A), low-invasive breast cancer cells (MCF7), metastatic breast cancer cells (MDA-MB-231), and aggressive prostate cancer cells (PC-3) (Fig. 1c), whose plasma membrane tensions were reported as 91.89 pN/$\mu$m, 82.78 pN/$\mu$m, 45.19 pN/$\mu$m (ruffling) (50.45 pN/$\mu$m blebbing), and 38.33 (ruffling) (42.61 pN/$\mu$m, blebbing)[7]. Among the cell types, MCF10A cells showed the highest average DTD counts, and MCF7 cells showed higher counts than did MDA-MB-231 and PC-3 cells. This observation highlights the low cell surface tension

in highly invasive and aggressive cancer cells and is consistent with the results of measurement of plasma membrane tension using optical tweezers[7].

## Cell surface mechanics related to cancer malignancy

We investigated whether ELASTomics allows joint analysis of cell surface tension and gene expression using four cell types (MCF10A, MCF7, MDA-MB-231, and PC-3) (Fig. 1f and Supplementary Fig. 7a), simultaneously profiling the mechanical and molecular phenotypes. To facilitate the identification of cell types in the scRNA-seq data analysis, we spiked 62-nucleotides long tags free of dextran into the DTD buffer as a cell-hashing tag[34], electroporated individual cell types with DTD, and ran 10x Genomics Single Cell 3′ v3.1 with a pool of DTD-labelled cells. We integrated data from six batches, pooling two or three different cell types or experimental conditions into a single dataset. We identified 3804 PC-3 cells (median of 10,805 UMI and 3432 genes detected per cell), 2275 MDA-MB-231 cells (median of 4594 UMI and 1924 genes detected per cell), 1575 MCF7 cells (median of 13,308 UMI and 3536 genes detected per cell), and 2083 MCF10A cells (median of 6124 UMI and 2052 genes detected per cell) with nanopore-electroporated cells. We also identified 4018 PC-3 cells (median of 10,800 UMI and 3440 genes detected per cell), 1691 MDA-MB-231 cells (median of 6613 UMI and 2494 genes detected per cell), 3087 MCF7 cells (median of 13,136 UMI and 3363 genes detected per cell), and 3412 MCF10A cells (median of 4306 UMI and 1531 genes detected per cell) with the control cells. As the comparison, we also identified 5072 PC-3 cells applied with 75 V (median of 8086 UMI and 2888 genes detected per cell) (Supplementary Fig. 3c-e), 180 MCF10A cells treated by cytochalasin D and applied with 40 V (median of 32,706 UMI and 5316 genes detected per cell) (Fig. 1b), and 822 MCF10A cells applied with 40 V (median of 14,350 UMI and 3534 genes detected per cell) (Fig. 1b and Supplementary Fig. 3s-u). Uniform manifold approximation and projection (UMAP) and Louvain community detection using Seurat (version 4.0.4) identified four segregated clusters of expressed marker genes and DTD counts for individual cell types (Fig. 1g-i and Supplementary Fig. 7b-e). The cell-hashing tag yielded consistent counts in each cluster, supporting the identification of cell types of clusters by gene expression (Supplementary Fig. 7f-i).

To identify genes that commonly contribute to the regulation of cell surface tension across different cell types, we computed the correlation coefficient between DTD abundance and gene expression using data from two human breast cancer cell lines, MDA-MB-231 and MCF7 (Fig. 1j). As expected, the genes with higher expression in MCF7 than in MDA-MB-231 had a positive correlation with DTD counts and vice versa, although with notable exceptions. For instance, the expression of ribosomal protein L11 (*RPL11*) and L37 (*RPL37*) showed an inverse correlation with DTD abundance, while the differences in expression between the two cell types were insignificant. Gene ontology (GO) enrichment analysis in positively (correlation >0.2) or negatively (correlation <−0.15) correlated genes showed enrichment in genes that regulate the cell adhesion, including cadherin binding and cell adhesion mediator activity (Supplementary Fig. 8a), and essential genes that regulate the cell surface mechanics, including actin binding, myosin binding (Supplementary Fig. 8b). To elucidate functional enrichment uniquely identified by the correlation with DTD abundance, we performed gene set enrichment analysis (GSEA) by ordering genes according to the correlation coefficient. Interestingly, GSEA based on the correlation with the DTD counts showed enrichment in some gene sets such as myosin heavy chain class II protein complex binding (GO:0023026) and symporter activity (GO:0015293), which were insignificant in a standard GSEA based on the fold change of gene expressions between the two cell types (Fig. 1k, l and Supplementary Fig. 8c), highlighting the shared enrichment in the cytoskeleton-related proteins among MCF7 and MDA-MB-231 cells. We note that

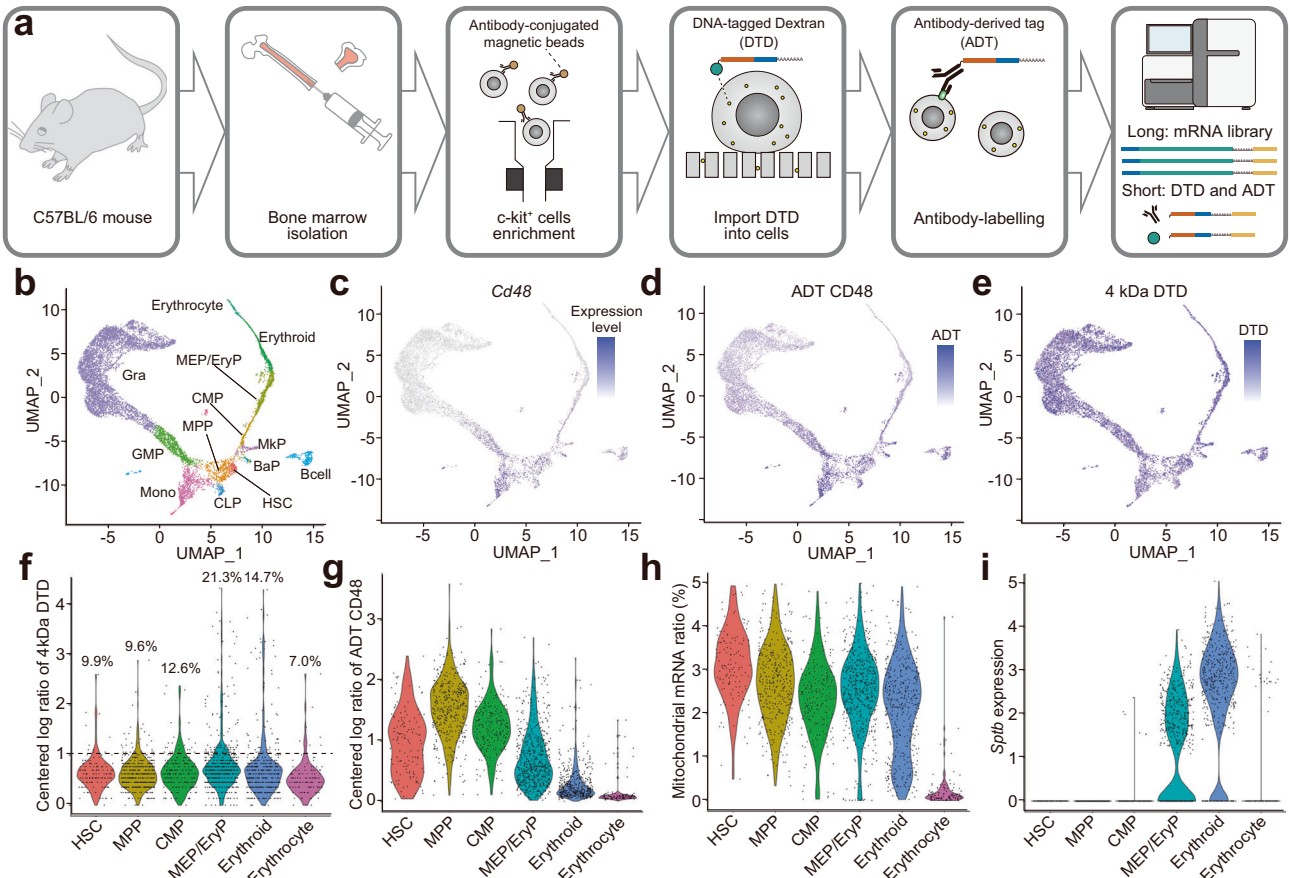

**Fig. 2 | ELASTomics analysis of mouse hematopoietic stem/progenitor cell (mHSPCs). a** Schematic image of ELASTomics with CITE-seq for mHSPCs. c-kit+ bone marrow (BM) cells were isolated from a C57BL/6 J mouse femur using antibody-conjugated magnetic beads. After DTDs were imported into the c-kit+ BM cells by nanopore-electroporation, the cells were incubated with antibodies conjugated with antibody-derived tag (ADT). The DTDs, ADTs, and mRNA expression were measured by scRNA-seq. **b–e** UMAP of mHSPCs. The color and color gradient indicates the identified cell types (Gra: granulocyte $n = 5786$; GMP: granulocyte-macrophage progenitor $n = 857$; Mono: monocyte $n = 730$; CLP: common lymphoid progenitor $n = 113$; MPP: multipotential progenitors $n = 394$; HSC: hematopoietic

stem cell $n = 131$; MkP: megakaryocyte progenitors $n = 126$; BaP: basophil progenitors $n = 69$; CMP: common myeloid progenitor $n = 206$; EryP/MEP, erythroid progenitors / megakaryocyte-erythrocyte progenitors $n = 573$; Erythroid $n = 463$; Erythrocyte $n = 115$; Bcell $n = 356$; other $n = 92$) (**b**), *Cd48* gene expression (**c**), centered log ratios of Cd48 ADT (**d**), and centered log ratio of 4 kDa DTD (**e**), respectively. **f–i** Comparison of the erythroid lineage cells (HSC, MPP, CMP, MEP/EryP, erythroid, and erythrocyte) with centered log ratios of 4 kDa DTD **f**, centered log ratios of Cd48 ADT (**g**), percent fractions of mitochondrial mRNA (**h**), and gene expressions of *Sptb* (**i**), respectively. **f** The percentage indicates the ratio of cells higher than 1 of the centered log ratio of the 4 kDa DTD.

gene expression in MCF10A cells were slightly perturbed by the nanopore-electroporation (Fig. 1g-i). GSEA showed enrichment in the cellular response to stress, cell death, and regulation of cell death (Supplementary Fig. 8d), consistent with a previous work[35].

## Cell surface mechanics during differentiation of haematopoietic cells

Cell differentiation, including the supply of mature blood cells from haematopoietic stem cells, usually occurs in parallel with the maturation of cell membranes to enable specialized functions. Therefore, we investigated whether ELASTomics can reveal the evolution of cell surface tension during normal haematopoietic cell differentiation in mouse bone marrow (BM) (Supplementary Fig. 9a). BM cells, including HSPCs, crudely isolated from normal C57BL/6 J mice (8 weeks) using a c-kit+ magnetic column (Fig. 2a), were used for analysis. To apply ELASTomics to primary non-adherent cells, which are challenging cell types, we allowed cells to settle on the track-etched membrane precoated with 100 μg/mL fibronectin for 1 h and then applied 75 V for nanopore electroporation. To assist identification of cell states, we labeled the nanopore-electroporated HSPCs with antibody-oligonucleotide conjugates, targeting three surface markers (Sca-1,

CD48, and CD150) after nanopore electroporation. We carried out the CITE-seq workflow, which yields a library (long) for mRNA-seq, as well as a library (short) for DTD and antibody-derived tags. We identified 10,011 mHSPCs with a median of 6893 UMI and 1919 genes detected per cell. The UMAP of mHSPCs showed a hierarchical and continuous change in cell states during differentiation, based on expression of marker genes and surface proteins (Fig. 2b-d and Supplementary Fig. 9b-o). The projection of DTD counts on the UMAP showed that nanopore electroporation enabled the import of DTD to the non-adherent cells in various states (Fig.2e).

We focused on ELASTomics data relating to erythroid differentiation from HSPCs, as structural changes in the cell membrane are essential during erythroid maturation and differentiation, including for denucleation[36]. ELASTomics data revealed that a notable fraction of megakaryocyte-erythrocyte progenitors/erythroid progenitors (MEP/EryP) and erythroids received more DTDs than did other cell types, implying that cells committed to erythrocyte differentiation transiently increase cell surface tension before denucleation at the erythrocyte stage (Fig. 2f-h). Additionally, we performed nanopore electroporation on mHSPCs using FITC-BSA and confirmed that Ter-119-positive cells, which are erythroid progenitor cells, had higher cell

surface tension than did Ter-119-negative cells (Supplementary Fig. 9p, q), supporting the findings of ELASTomics.

To explore genes that were coupled with changes in cell surface tension during cell differentiation, we calculated the correlation coefficient between DTD abundance and gene expression in the erythrocyte lineage (haematopoietic stem cell (HSC)–multipotential progenitor (MPP)–common myeloid progenitor (CMP)–MEP/EryP–erythroid–erythrocyte) (Supplementary Fig. 9r, s). To overcome relatively low DTD counts in mHSPCs, we filtered out cells with less than 1 centred log ratio of DTD in calculating the correlation. The data identified 54 genes with expression levels with a positive correlation with DTD counts (Pearson's $r > 0.2$ and adjusted $P < 10^{-4}$) and underscored the involvement of spectrin alpha 1 (*Spta1*) and spectrin beta (*Sptb*), genes related to the regulation of erythroid membrane structure, in the transient increase in cell surface tension (Fig. 2i). GSEA also showed enrichment in spectrin binding (GO:0030507) and gene sets which related to the regulation of spectrin via ATP-dependent phospholipid flippase[37] (Supplementary Fig. 10). In contrast, in the lineage from mHSPCs to granulocytes, only one gene showed a significant correlation of expression with cell surface tension (Supplementary Fig. 9t, u).

### Cell surface mechanics during cellular senescence

Senescent cells exhibit changes in mechanical properties, undergoing an increase in stiffness or tensile force in parallel with a reduction in elasticity and strength[38]. Alterations in cellular mechanical properties lead to progressive dysregulation of mechanosensitive signaling[39,40] and the cytoskeleton[41]. Therefore, we investigated whether multimodal data from ELASTomics could enhance an understanding of surface mechanics changes during cellular scenescence[9]. We performed ELASTomics with human foetal lung fibroblasts (TIG-1), which exhibit replicative senescence accompanied by changes in cholesterol abundance in lipid rafts[42], chromosomal instability[43], abnormal glycation and Golgi transport[44], abnormal lipid accumulation[45], and mitochondrial dysfunction[46] as cellular senescence progresses, by applying 50 V pulses. We identified 4654 TIG-1 cells (applied voltage: 0 V) with a median of 13,130 UMI and 3648 genes and 4711 TIG-1 cells (applied voltage: 50 V) with a median of 19,207 UMI and 4474 genes detected per cell. Cells with senescence signatures were more abundant in a relatively senescent TIG-1 population (population doubling level (PDL) = 50–60) than in a young TIG-1 population (PDL = 30–40) at the population level (Supplementary Fig. 11a-c), while the expression levels of various senescence-associated genes in individual TIG-1 cells were heterogeneous at a single-cell level (Supplementary Fig. 12a). Consistent with this, ELASTomics revealed that, at a population level, the number of TIG-1 cells with high cell surface tension was higher in the senescent TIG-1 population than in the young TIG-1 population (Fig. 3a, b). We also confirmed that on average the surface tension of senescent TIG-1 cells was higher than that of young TIG-1 cells by AFM (Supplementary Fig. 11d). Further, the surface tension of individual cells was heterogeneous (Fig. 3b-e), consistent with heterogeneous expression of senescence-associated genes (Supplementary Fig. 12b). To explore genes that were coupled with the changes in cell surface tension during cellular senescence, we computed the correlation coefficient between DTD abundance and gene expression (Fig. 3f and Supplementary Fig. 12c-f). We observed that expression levels of genes up-regulated during senescence were positively correlated with DTD abundance, including for Fos proto-oncogene (*FOS*), adrenomedullin (*ADM*)[47], and cyclin-dependent kinase inhibitor 1 A (*CDKN1A*)[48]. We found that expression levels of genes down-regulated during senescence were negatively correlated with DTD abundance, for example for deoxythymidylate kinase (*DTYMK*) and disks large-associated protein 5 (*DLGAP5*)[49]. Further, expression levels for genes involved in DNA replication and repair showed a negative correlation with DTD abundance, for example for nucleolin (*NCL*) and clamp-associated factor (*PCLAF*), concordant with the decline in cell proliferation during cellular senescence.

To identify genes that regulate cell surface tension during senescence, we filtered 210 genes out of the genes detected by thresholding at Pearson's $r > 0.15$ and $P < 10^{-10}$ and identified Ras-related glycolysis inhibitor and calcium channel regulator gene (*RRAD*) as a potential regulator by regression using the elastic-net generalized linear model in which *RRAD* showed the fourth highest effect on cell surface tension following two long-non-coding RNAs, *AC007952.4* and *AC091271.1*, and *KLF2* genes (Fig. 3g). Additionally, we performed bulk RNA-seq and nanopore electroporation for TIG-1 cells at different PDL values and analysed gene expression and the amount of imported FITC-BSA (Fig. 3h-j and Supplementary Fig. 11e-m). In line with the findings of ELASTomics, the population of TIG-1 cells increased both the amount of imported FITC-BSA and the expression of *RRAD* as PDL increased while not *KLF2*, indicating that cell surface tension and *RRAD* expression are related during cellular senescence. *RRAD* is a biomarker up-regulated by senescence-induced stimuli and its expression increases with age in human skin and adipose tissues[50]. *RRAD* represses glycolysis mainly through inhibition of glucose transporter-1 (GLUT1) translocation to the plasma membrane[51]. Expression of *RRAD* induces p53-mediated inhibition of lung carcinoma cell migration[52], and hypermethylation correlates with poor prognosis in lung adenocarcinomas[53]. We thus hypothesized that *RRAD* plays a crucial role in regulating cell surface tension and coupled functional changes during the senescence of TIG-1 cells via the glycolysis pathway. GSEA also showed enrichment in glycolysis-related gene sets such as ATP-dependent activity (GO:0140657), and ATP hydrolysis activity (GO:0016887) as well as aging (GO:0007568) (Supplementary Fig. 12g-k).

To investigate the causal relationship between the expression of *RRAD* and cell surface tension, we perturbed expression of *RRAD* using a small interfering RNA (Supplementary Fig. 11n). Suppression of *RRAD* expression markedly recovered cell surface tension in TIG-1 cells at PDL values of 40 and 56 compared to cell surface tension for negative controls (Fig. 3k). In addition, we have confirmed that inhibition of glycolysis with 2-deoxy-D-glucose (2-DG) increases the cell surface tension of TIG-1 cells. (Fig. 3l). The results indicate that *RRAD* contributes to an increase in cell surface tension during senescence via glycolysis in TIG-1 cells. Our multimodal approach uniquely enabled us to identify *RRAD* as a senescence-related gene connected to cell surface mechanics.

## Discussion

Here, we report ELASTomics, a method that integrates profiling of cell surface mechanics with unbiased transcriptomic analyses of thousands of single cells. ELASTomics enables the measurement of cell surface tension by importing DTD into cells using nanopore electroporation and couples quantification of DTD abundance with gene expression information obtained using scRNA-seq. ELASTomics provides a comprehensive view of gene expression underlying changes in the surface tension of cells in various biological contexts and allows us to discover important genes that regulate cell surface tension. In this study, we demonstrate the application of ELASTomics using cancer cells from various types of malignancies, primary HSPCs isolated from the BM of mice, and TIG-1 cells at different senescence levels. ELAS-Tomics has the flexibility to tune sensitivity by controlling the magnitude of the electric field in cases where the electric field does not compromise gene expression in cells and thus is applicable to a broad range of cell types, including suspended cells and non-mammalian cells.

ELASTomics data on MCF7 and MDA-MB-231 cells identified the shared negative correlation of *RPL11* and *RPL37* expressions with DTD counts (Fig. 1j). *RPL37* regulates the expression level of p53, which is a tumour suppresser, by binding to Mdm2, a RING-type E3 ubiquitin

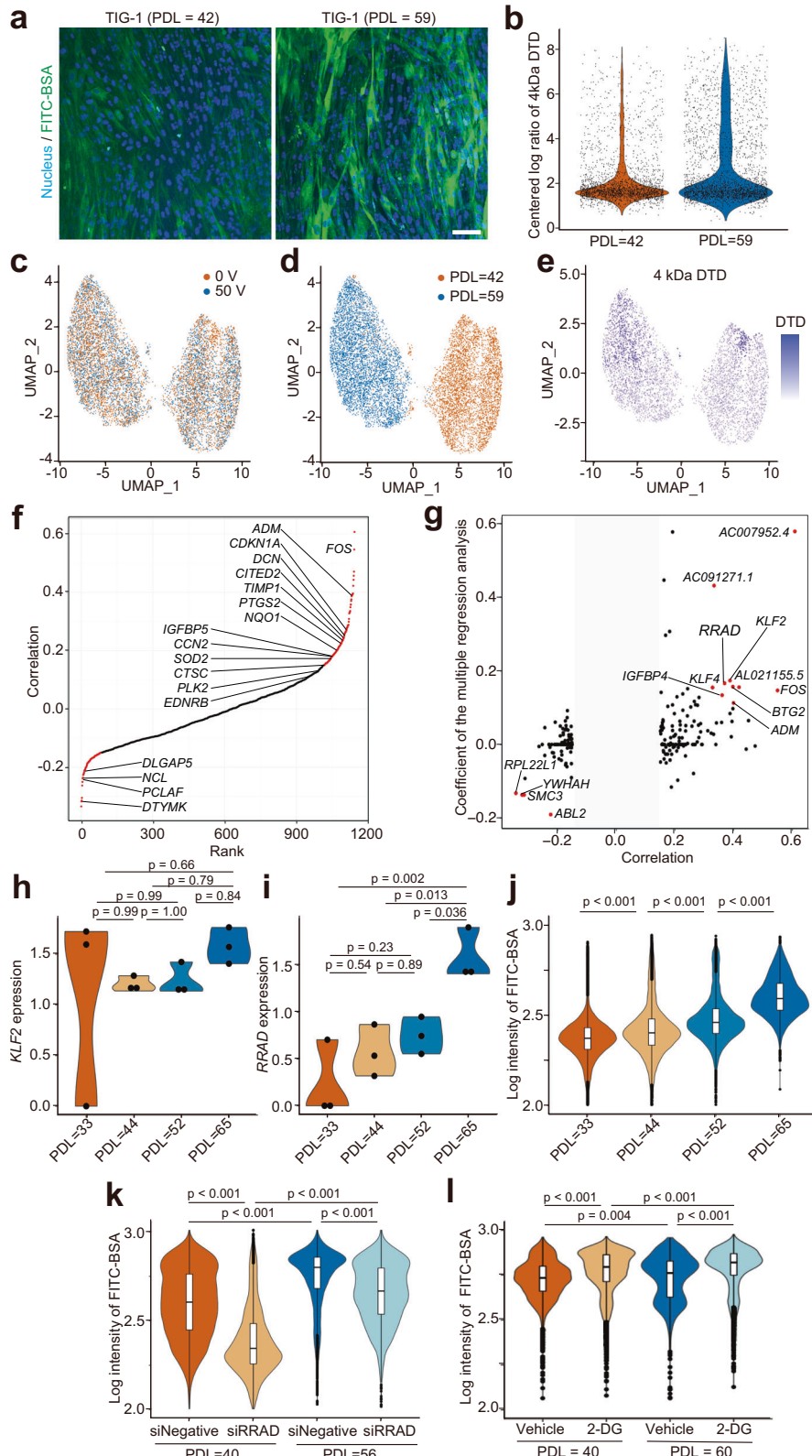

ligase[54,55]. RPL11 also binds to Mdm2 to inhibit the ubiquitination and degradation of p53[56]. Taken together, the negative correlation of *RPL11* and *RPL37* with DTD counts may imply that the regulation of the cell surface tension via the p53 axis is conserved among MCF7 and MDA-MB-231 independent of the cancer cell type. GSEA showed enrichment not only for cytoskeleton-related gene sets such as MHC class II protein complex binding but also for symporter activity and sodium ion

transmembrane transporter activity (Supplementary Fig. 8c). Changes in intracellular ion concentrations induce changes in osmotic pressure and cell surface tension[31,57]. Thus, ionic channels and transporters may contribute to changes in cell surface tension by transporting intracellular ions to alter osmolarity[35].

Cell surface epitopes are reliable indicators for identifying cell states in a complex population. CITE-seq combines the detection of

**Fig. 3 | ELASTomics analysis of cellular senescence. a** Fluorescence images of nanopore-electroporated TIG-1 cells at different population doubling levels (PDL) = 42 and 59. FITC-BSA (green) was imported into cells by nanopore-electroporation. Nuclei were stained by Hoechst 33342 (blue). Similar results were observed in independent duplicate experiments. Scale bar, 50 μm. **b** Centered log ratio of 4 kDa DTD in TIG-1 cells at PDL = 42 and 59. Dots indicate individual cells. **c-e** UMAP of relatively young (PDL = 42) and senescent (PDL = 59) TIG-1 cells. The colors respectively indicate the applied voltages (**c**), PDLs (**d**), and centered log ratio of 4 kDa DTD (**e**). **f** The coefficients of correlation between the centered log ratio of 4 kDa DTD and expression of individual genes. Red points are the genes with Pearson's $|r| > 0.15$. **g** The coefficients of variables (s) and the coefficients of correlation between DTD counts and gene expression. Red points are genes with Pearson's $r > 0.3$ or $< -0.2$ ($P < 0.001$) and $|s| > 0.1$. **h, i** Comparison of bulk TIG-1 cells at different PDLs (PDL = 33, 44, 52, and 65) with the expression levels of *KLF2*

(**h**) and *RRAD* (**i**) ($n = 3$ biologically independent samples). **J-l** Fluorescence intensity of imported FITC-BSA by nanopore-electroporation to TIG-1 cells with different PDLs (PDL = 33 $n = 11,008$; PDL = 44 $n = 11,050$; PDL = 52 $n = 11,171$; PDL = 65 $n = 9365$ independent cells) (**j**), with or without the suppression of *RRAD* by siRNA (PDL = 40 with siNegative $n = 9711$; PDL = 40 with siRRAD $n = 11,012$; PDL = 56 with siNegative $n = 10,636$; PDL = 56 with siRRAD $n = 4894$ independent cells) (**k**), and treated with vehicle or 2-deoxy-D-glucose (2-DG) (PDL = 40 with 0 mM 2-DG $n = 10,931$; PDL = 40 with 1 mM 2-DG $n = 11,076$; PDL = 60 with 0 mM 2-DG $n = 2861$; PDL = 60 with 1 mM 2-DG $n = 6083$ independent cells) (**l**). In the boxplots (**j–l**), the center lines are the median, the box indicates the first and third quartiles, whiskers are the minimum and the maximum excluding outliers, and dots are outliers. Similar results were observed in independent duplicate experiments. The P values (*p*) are indicated in the graph (**h-l**: Tukey's t-test). Source data are provided as a Source Data file.

cell surface epitopes with unbiased transcriptomic profiling, and ELASTomics can be integrated with the CITE-seq workflow. We demonstrated the feasibility of integrating ELASTomics and CITE-seq, and profiled cell states along cell lineages of mHSPCs via molecular and mechanical phenotyping. The multimodal dataset revealed that erythroid-lineage progenitor cells transiently increase cell surface tension during lineage commitment and maturation of erythrocytes. Further, analysis of the correlation between the cell surface tension and gene expression profiles revealed that *Spta1* and *Sptb*, components of the spectrin tetramer, are involved in the transient change in cell surface tension during cell differentiation (Fig. 2f, i). Since the spectrin tetramer is a primary structural member of the membrane skeleton for stability and deformability in erythrocytes[58–60], it is reasonable that the expression of spectrin genes is up-regulated in parallel with an elevation of cell surface tension during maturation of erythroid cells. The results suggest that ELASTomics uniquely illuminates cell surface mechanics and coupled gene regulation during differentiation of mHSPCs.

ELASTomics represents a powerful approach for discovering regulatory genes that contribute to the surface tension of single cells. The ELASTomics dataset for TIG-1 cells at different senescence levels revealed that cell surface tension increased during replicative senescence. The analysis detected genes related to cellular senescence, including *FOS*, *CDKN1A*, and *PCLAF*, by positive or negative correlation between gene expression and DTD abundance. Further, regression analysis identified *RRAD* as a potential regulator of cell surface tension during senescence in TIG-1 cells. We validated the impact of *RRAD* expression on cell surface tension by silencing *RRAD* and observing recovery in the cell surface tension of senescent TIG-1 cells. *RRAD* is known to repress glycolysis, inhibiting GLUT1 translocation to the plasma membrane[51]. Based on our findings, cell surface tension during cellular senescence may increase with the suppression of glycolysis via up-regulation of RRAD[61], though further work is needed to elucidate the complete molecular pathway. TIG-1 cell was isolated from a human fetal lung for the study of cellular senescence[62]. TIG-1 cells have also been utilised to study ageing-related phenomena, including mitochondrial dysfunction, excessive reactive oxygen species production[46], and abnormal lipid accumulation[45]. An analysis utilised TIG-1 cell identified that *ATP6VOA2*, a gene responsible for autosomal recessive cutis laxa type 2 (ARCL2), triggers abnormal glycosylation and Golgi transport as a cellular senescence program[44]. Accordingly, owing to the low genetic variation and high reproducibility, TIG-1 cell is advantageous when screening specific genes involved in cell surface tension along cellular senescence. We note that although TIG-1 cells are a well-established cell line for studying replicative senescence[44,46,63], the cultured senescent cells, including TIG-1 cells behave somewhat differently than senescent cells in vivo[64]. We envision that ELASTomics would be applicable to the primary fibroblasts by integrating with rapid cell isolation protocols[65,66]. As previously reported, the regulation of cytoskeleton is also highly related to the glycolysis in cancer cells, which maintain high glycolytic rates[61]. *RRAD* expression is

down-regulated by DNA methylation in malignant lung and breast cancers[53]. It has also been reported that *TXNIP*, which inhibits the translocation of GLUT1 to the plasma membrane like *RRAD*, alters cell mechanics in some cancer cells[67]. Indeed, we found a positive correlation between cell surface tension and *TXNIP* expression in cancer cell lines (Supplementary Fig. 13a, b). We thus envision that inhibition of the glycolysis pathway, including upregulation of *RRAD*, may play a general role in the regulation of cell surface tension in various contexts. We note that although we have examined the correlation between the *RRAD* expression and the cell surface tension with MCF7, MDA-MB-231, and PC-3 cells, we could not confirm the correlation in our current dataset owing to the low sensitivity of scRNA-seq (Supplementary Fig. 13c).

Although ELASTomics is a versatile approach capable of assaying various cell types, including non-adherent cells, the magnitude of the electric field must be adjusted to obtain the optimum signal-to-noise ratio (SNR) while minimising perturbations of gene expression. For instance, for best outcomes we treated mHSPCs, which are non-adherent cells, with 75 V for reliable DTD counting but used 40 or 50 V for adherent cells. This implies that cell adhesion to the substrate influences nanopore electroporation and DTD import, which affects SNR in DTD counting. Therefore, when assaying a highly heterogeneous cell population with various adhesion modes, DTD abundance reflects other cellular states as well as cell surface tension. The adhesion area of a cell on the track-etched membrane is another variable that influences DTD abundance, because the amount of DTD imported into a cell depends on the number of pores. This variability can be normalized by using DTD with a range of Stokes radii (Supplementary Fig. 6a-c)[31].

In summary, we present a method, ELASTomics, that captures the landscape of cell surface mechanics, as well as the underlying transcriptional regulation and enables the identification of key genes involved in the regulation of cell surface tension. We envision that ELASTomics will provide important insights into cell surface mechanics in broad biological contexts, including cancer biology, tissue development, and cellular senescence.

## Methods
### Ethical statement
Ethical approval of this study protocol for animal experiments was obtained from RIKEN (authorization number W2021-200079). The animal facilities were maintained at 20–25 °C with 40–60% humidity under a standard 12-h light–dark cycle.

### Cell
MCF10A human non-tumorigenic mammary epithelial cells (CRL-10317, ATCC) were maintained in DMEM/F-12 (08460-95, Nacalai Tesque) supplemented with 10% fetal bovine serum (FBS) (26140-079, gibco), 20 ng/mL epidermal growth factor (059-07873, FUJIFILM), 10 μg/mL insulin (19278-5 ML, Sigma-Aldrich), 0.5 μg/mL hydrocortisone (086-10191, FUJIFILM), and 1% penicillin-streptomycin (P/S)

(P4333-100ML, Sigma-Aldrich). MCF7 human breast cancer cells (HTB-22, ATCC) were maintained in MEM (M4655-500ML, Sigma-Aldrich) or DMEM (08456-65, Nacalai Tesque) with 10% FBS and 1% P/S. PC-3 human prostate cancer cell (RCB2145, RIKEN BRC), OVCAR-3 human ovary cancer cells (RCB2135, RIKEN BRC), and K562 non-adherent cancer cell (RCB0027, RIKEN BRC) were maintained in RPMI-1640 medium (30264-85, Nacalai Tesque) with 10% FBS and 1% P/S. MDA-MB-231 human breast cancer cells (HTB-26, ATCC), TIG-1 human lung-derived cells (JCRB0501 and JCRB0504, JCRB), and HeLa human cervical cancer cells (RCB0007, RIKEN BRC) were respectively maintained in DMEM (08456-65, Nacalai Tesque) containing 10% FBS and 1% P/S. CHO-K1 cells (RCB0285, RIKEN BRC) derived from ovary of Chinese hamster were maintained in Ham's F-12 medium (087-08335, FUJIFILM) supplemented with 10% FBS. Mammalian cell lines were cultured at 37˚C in 5% $CO_2$. GEM-81 cells (RCB1174, RIKEN BRC) derived from the goldfish were maintained in Leibovitz's L-15 medium supplemented with 2 mM L-glutamine, 20% FBS, and 1% P/S at 25 °C. We regularly tested for mycoplasma contamination using a CycleavePCR™ Mycoplasma Detection Kit (CY232, TAKARA).

## Mouse bone marrow cells

All experimental procedures were performed in accordance with the guideline of the Animal Experiment Committee of RIKEN. Mouse bone marrow (BM) cells were isolated from femurs of 8-week-old male C57BL/6 J mice (JSLC Co., Hamamatsu, Japan) according to an established protocol[68]. Briefly, BM cells were suspended in the 50 μL PBS(-), added 10 μL c-kit (CD117) microbeads (130-097-146, Miltenyi Biotec), and then collected by LS column (130-042-401, Miltenyi Biotec). c-kit⁺ BM cells were washed with PBS (-).

## RNA interference

RRAD Silencer Select Pre-designed siRNA (s12348, Thermo Fisher Scientific) or Silencer Negative Control No.1 siRNA (AM4611, Thermo Fisher Scientific) were used for the RNA interference. 5 pmol siRNA and 1.5 μL Lipofectamine RNAiMAX (13778030, Thermo Fisher Scientific) were diluted in 50 μL Opti-MEM (Gibco) and incubated at room temperature for 5 min. TIG-1 cell ($2.5× 10^4$ cells) was incubated with 100 μL siRNA mixtures at 37˚C in 5% $CO_2$ for 1-3 days and then used for experiments.

## Perturbation of cell surface mechanics

To reduce the cortical tension by disrupting the regulation of the actin cytoskeleton, cells were incubated with 10 μM cytochalasin D (037-17561, FUJIFILM), (-)-Blebbistatin (021-17041, FUJIFILM), or Y-27632 (10005583, CAYMAN) in PBS for 120 minutes at room temperature. To change the plasma membrane tension, cholesterol in the plasma membrane was removed by methyl-β-cyclodextrin (MβCD). 250 mM MβCD (332615, Sigma-Aldrich) in Milli-Q water was rotated with or without 50 mM cholesterol (C8667, Sigma-Aldrich) for 1 h, diluted to 10 mM with serum-free DMEM, and then filtered through 0.22 μm Millex-GP (SLGPR33RS, Merck). Cells were incubated with DMEM containing 10 mM MβCD or MβCD/cholesterol for 1 h at 37˚C.

To inhibit the glycolysis, cells were incubated with 1 mM 2-Deoxy-D-glucose (10722-11, Nacalai Tesque) in DMEM for 1 day at 37˚C in 5% $CO_2$.

## Measurement of cellular senescence

To compare the senescence phenotype, TIG-1 cells were stained with DNA damage detection kit (G266, Dojindo). After fixing and staining with anti-γH2AX primary antibody (G266, 1:50, Dojindo), cells were stained with Hoechst33342 and secondary antibody-Red (G266, 1:50, Dojindo) and then imaged using a FV1000-D confocal microscope (OLYMPUS) to calculate the fraction of γH2AX-positive cells. To perform real-time RT-PCR, 500 TIG-1 cells with different PDLs were lysed in 50 μL lysis buffer, including 5 μL 10 x Lysis buffer (Takara) and 0.5 μL

RNase inhibitor (40 unit/μL; Y9240L, enzymatics) and then heated at 72 ˚C for 3 min and immediately placed on ice for 2 min to denature RNA. 2 μL cell lysate was amplified using TaqMan® RNA-to-Ct™ 1-Step Kit (4392938, Thermo Fisher Scientific) and respective gene-specific 20× TaqMan gene expression assays as follows: (*ACTB*) Hs01060665_g1; (*GAPDH*) Hs04420697_g1; (*RRAD*) Hs00188163_m1; (*CDKN1A*) Hs99999142_m1; (*TIMP1*) Hs01092511_m1; and (*DCN*) Hs00754870_s1 (Thermo Fisher Scientific). The relative mRNA expression was calculated by normalizing the expression with *GAPDH* expression.

## Fabrication and device assembly for nanopore-electroporation

The device for nanopore-electroporation consists of two parts, the cell culture chamber and the electrode holder (Supplementary Fig. 2a-e). The cell culture chamber was made of a polydimethylsiloxane (PDMS) slab with a track-etched membrane on the bottom. A desktop 3D-printer (Form 3, Formlabs) was used to fabricate the mold of the cell culture chamber. The computer-aided design of the mold was made available as supplementary data. High Temp Resin V2 (Formlabs), which is tolerant up to 238˚C, was used to produce the mold. To remove uncured resin, freshly printed molds were cleaned by flushing and bathing in isopropanol (IPA) for 6 min, followed by air drying. The molds were post-cured at 80˚C for 120 min using a Form Cure (Formlabs). An additional thermal cure was performed in an oven at 160˚C for 180 min. PDMS (SILPOT 184, Dow Corning) mixture at a 10 (base material): 1 (curing agent) ratio was cast on the mold and then cured in an oven at 70˚C for 2 h. To fabricate the culture chamber, a hole with an 8-mm diameter was punched at the center of the PDMS slab removed from mold (Supplementary Fig. 2b). Track-etched membranes (Isopore, VCTP04700, Merck; pore size: 100 nm; pore density: $6.04 ± 0.52 \, \mu m^{-2}$) were cut to the same size as the bottom of the PDMS slab and glued on the bottom of the PDMS slab using the uncured PDMS (Supplementary Fig. 2c). To fix the track-etched membrane, the assembled PDMS slabs were cured in an oven at 70˚C for 30 min. The cell culture chambers were sterilized by a UV exposure for more than 10 min in a safety cabinet before use (Supplementary Fig. 2d).

To fabricate the electrode holder, the electrode holder was directly printed using the 3D printer with the same procedure as described above (Supplementary Fig. 2e). A platinum electrode (PT-356320, Nilaco) was glued on the bottom of the electrode holder with power cable using the pre-cured PDMS and fixed in an oven at 70˚C for 30 min.

## Electron microscopy images

For electron microscopy images, cells were fixed with 2.5% glutaraldehyde in 20 mM HEPES buffer (pH7.0) for 2 h. Samples were processed by 5% HILEM IL1000 (Hitachi High-Tech) at room temperature for 30 min, washed with diluted water, and then dried by air spray. Samples were attached with a carbon double-stick tape (Nisshin-EM) on a sample stage and coated with osmium tetroxide by an osmium coater (HPC-1SW; Vacuum device). Images were acquired using a field emission scanning electron microscope (SU8220; Hitachi High-Tech), working at an accelerating voltage of 2.0 kV with a secondary electron detector.

## Synthesis of DTD

We designed the oligonucleotide for DTD as the similar design of antibody-oligo sequences for CITE-seq[28]. Briefly, we purchased oligonucleotides with 5'-amino-modifier (AmC6) that contained standard TrueSeq small RNA read 2 sequences, an 8 nt long barcode for the identification of dextran, a 10 nt long unique molecular identifier (UMI), and a 32 nt long poly-As (Supplementary Table 2). To conjugate the AmC6 oligonucleotide with lysine-dextran, the AmC6 oligonucleotides were dried under nitrogen reflux and reacted with 200 equivalents of disuccinimidyl suberate (DSS) crosslinker (A39267,

Thermo Fisher Scientific) under 100 μL anhydrous dimethyl sulfoxide solvent (276855-100 ML, Sigma-Aldrich) containing 0.5% triethylamine (34805-82, Nacalai Tesque) at 60 °C for 1 day. 0.4 M sodium acetate (300 μL) and pure ethanol (1 mL) were added on ice for ethanol precipitation. After centrifugation at 14,000 x g for 1 h, pellet including DSS-conjugated oligonucleotides dried under nitrogen reflux. DSS-conjugated oligonucleotides with different barcodes were reacted respectively with 4, 10, 40, 70, 150, and 500 kDa FITC-lysine-dextran or Antonia Red™-lysine-dextran (FLD004, FLD010, ARLD040, FLD70, ARLD150, and FLD500, TdB Labs) under anhydrous DMSO solvent at 60 °C for 1 day. The synthesized DTDs were separated from unreacted oligonucleotides and lysine-dextrans by a 1.5% agarose gel electrophoresis. Stokes radii were measured by ELSZ-2PL zeta-potential & particle size analyzer (OTSUKA ELECTRONICS) (Supplementary Table 1). The conjugation was assessed by gel electrophoresis using 0.8-3.0% agarose gel (Supplementary Fig. 1b). The concentration of DTD was quantified by reverse transcription (RT) followed by real-time PCR.

We used a previously reported microfluidic approach[69] to measure electrophoretic mobilities. We used HEPES-based buffer as the running buffer consisting of 20 mM HEPES and 260 mM sucrose (pH 7.0, resistivity 35.1 Ωm). We mixed each DTDs or dextrans and rhodamine B in the running buffer and introduced it into a cross-patterned microchannel (NS12A, Caliper Life Sciences) filled with the running buffer. We applied voltage to the channel via a computer-controlled voltage source (HVS448-3000D-LC, Labsmith), leading to the migration and separation of the analytes and rhodamine B in a microchannel branch (45.59 mm from cross junction to outlet) during current measurement (Keithley 6487 Picoammeter Voltage Source, Keithley Instruments). We observed the migration via epi-microscopy at 5 mm (20 mm for 150-kDa DTD) downstream from the cross junction. We fit the fluorescence intensities by Gaussian models using the 'fit' function in MATLAB and estimate the migration time of the analytes ($t_{migration,analyte}$) and rhodamine B ($t_{migration,RB}$) from the peak. We calculated the electrophoretic mobility of the analyte ($μ_{analyte}$) from the equation below.

$$μ_{analyte} = \frac{AL}{ρI}\left(\frac{1}{t_{migration,analyte}} - \frac{1}{t_{migration,RB}}\right) \quad (1)$$

Here, $A$ is the cross-sectional area of the microchannel, $I$ is the current of the separation channel, $L$ is the distance from the cross junction to the observation point, and $ρ$ is the buffer resistivity.

## Labeling cells with DTD via nanopore-electroporation

The cells were cultured in the cell culture chamber precoated with 100 μg/mL fibronectin (063-05591, FUJIFILM) at 37 °C for 1 day or 1 h (mHPSCs and K562 cells). After replacing the medium in the cell culture chamber with PBS (14249-24, Nacalai tesque), the bottom of the cell culture chamber was briefly immersed in a HEPES-based buffer (20 mM HEPES/NaOH pH 7.0, 260 mM sucrose) containing 0.1% pluronic F-127 (P2443-250G, Sigma-Aldrich) to prevent nonspecific adsorption of DTDs. The cell culture chamber was then placed on the electrode holder prefilled with 150 μL of a HEPES-based buffer, including DTD and cell hashtags at the concentration summarised in the Supplementary Table 3. DTD concentration is an important parameter that influences the number of DTD counts: a higher concentration yields more DTD counts. To gain the maximum signal-to-noise ratio, we here used the maximum concentration that we could prepare, depending on the type of DTD, as summarised in Supplementary Table 3. In the experiments for flow cytometry and fluorescence microscopy, we mixed 200 μg/mL FITC-BSA into the HEPES-based buffer to quantify the amount of the imported molecule via fluorescence measurement. The second platinum electrode was immersed in the PBS in the culture chamber. After incubating for

3 min, square pulses of 5 ms were applied between the pair of platinum electrodes for 500 cycles using a multifunction generator (WF1947, NF Corporation) and a high-speed bipolar amplifier (HSA4014, NF Corporation). The applied voltage was 40 V for cancer cell lines, 75 V for mHSPCs, and 50 V for TIG-1 cells, as determined from the FITC-BSA experiments. The applied voltages and currents were monitored with an in-house electric circuit controlled by a python program. After 2 min, the cell culture chamber was transferred to a six-well plate prefilled with the culture medium, and then PBS in the culture chamber was replaced with a fresh culture medium. Cells were incubated for 30 min at 37 °C in 5% CO$_2$. The adherent cells were detached from the track-etched membrane with trypsin (35555-54, Nacalai tesque) and washed three times with PBS, including 0.1% bovine serum albumin (BSA) (AM2616, Sigma-Aldrich) via centrifugation at 390 g for 3 min. The cells filtered with 40 μm pluriStrainer-Mini (43-10040, pluriSelect) were processed with Chromium Next GEM Single Cell 3′ Kit v3.1 (10x genomics).

## Labeling cells with oligonucleotide conjugated antibodies for cellular indexing of transcriptomes and epitopes by sequencing (CITE-seq)

In the experiments with mHSPCs, we labeled the nanopore-electroporated cells with TotalSeq™-A0130 anti-mouse Ly-6A/E (Sca-1) Antibody (10814, Biolegend), TotalSeq™-A0012 anti-mouse CD117 (c-kit) Antibody (105843, Biolegend), TotalSeq™-A0203 anti-mouse CD150 (SLAM) Antibody (115945, Biolegend), and TotalSeq™-A0429 anti-mouse CD48 Antibody (103447, Biolegend). After nanopore-electroporation, mHPSCs were incubated in 50 μL PBS (0.1% BSA), including 10 ng/μL TotalSeq antibodies on ice for 30 min and washed with PBS (0.1% BSA).

## Library construction for ELASTomics

We used the protocol of TotalSeq™-A (BioLegend, https://www.biolegend.com/ja-jp/protocols/totalseq-a-antibodies-and-cell-hashing-with-10x-single-cell-3-reagent-kit-v3-3-1-protocol) to construct the transcriptome and DTD libraries with some modifications. The nanopore-electroporated cells were suspended in PBS with 0.04% BSA and loaded onto the 10x Chromium Single Cell Platform (10x Genomics) at a concentration of 800–1000 cells/μL to generate GEMs as described in the manufacturer's protocol. To amplify cDNA derived from DTD, we spiked 3 nM of an additive primer (Supplementary Table 2) in the PCR reaction. Following the PCR, 0.6x SPRIselect (Beckman Coulter, Inc) was used to separate the long cDNA fraction derived from cellular mRNAs (retained on beads) against the short cDNA fraction derived from DTD and ADT (in the supernatant). The long cDNA fraction was processed according to the protocol to generate the transcriptome library. To capture short cDNA derived from DTD in the supernatant, 1.4x volume of SPRI beads was added to the supernatant, and processed, yielding an 11 μL of elution buffer. To construct the DTD library (including ADT when integrated with CITE-seq), the short cDNA diluted at 10-fold (2 μL) was amplified in 25 μL of 1xKAPA Hifi Hotstart Ready Mix (Roche) containing 0.5 μM of indexing primers (Supplementary Table 2) using the following program: 98 °C for 2 min; 2 cycles of 98 °C for 20 s and 74 °C for 30 s; 12 cycles of 98 °C for 20 sec and 72 °C for 30 sec; 72 °C for 5 min. Finally, the DTD libraries were cleaned using AMPureXP beads (1.5x) (Beckman Coulter) and eluted with 12 μL of elution buffer.

## Measurement of surface tension of individual cells by atomic force microscopy (AFM)

To analyze the relationship between the cell surface tension and the amount of imported molecules by nanopore-electroporation, we used AFM (Supplementary Fig. 4e-g)[30] and an epi-fluorescence microscopy. We here used transparent track-etched membranes (ipCELLCULTURE, 2000M23/620N403/13, it4ip; pore size: 400 nm; pore density: 2.0 ×10$^{-2}$ μm$^{-2}$) precoated with 100 μg/mL FN (063-05591, FUJIFILM) to

allow the fluorescence measurements. MCF10A cells were incubated in a culture chamber for 1 day at 37°C in 5% $CO_2$. After nanopore-electroporation, the amount of imported FITC-BSA was imaged by fluorescence microscopy. The fluorescence intensity of each cell was normalized by the average fluorescence intensity of all cells on the track-etched membrane using imageJ. To measure the cell surface tension of nanopore-electroporated cells, the track-etched membrane was peeled off from the PDMS slab and fixed on a glass slide. Cells were indented with a tall V-shaped and gold-coated BioLever (BL-RC150VB-C1, Olympus) (nominal spring constant $k = 0.006$ N/m). A spring constant, $k$, of a cantilever, was measured by the thermal method in water and was used to calculate the external force $P$ applied to the cell. Measurements were performed in PBS (-) at room temperature. To derive the surface tension $\sigma$, the force curve that showed the relation between external force $P$ and an indent depth $d$ was fitted with

$$P = \frac{2}{\pi} E^* d^2 \tan\phi \left[ 1 + \alpha_C \left( \frac{2\sigma/E^*}{d\tan\phi} \right)^{\beta_C} \right] \tag{2}$$

where $E^*$ presents the apparent elastic modulus, and $\phi$ is a half angle of conical tip. The tip of the cantilever was assumed as a point-symmetrical conical shape with a half angle of 45 degrees, and fitting parameters $\alpha_c$ and $\beta_c$ were 0.88 and 0.87, respectively.

### Flow cytometry and fluorescence microscopy of nanopore-electroporated cells
To detect live/dead cells, cells were incubated for 10 min in PBS containing 5 μg/mL propidium iodide (PI) and 5 μg/mL Hoechst 33342 at room temperature. mHPSCs were incubated in 50 μL PBS (0.1% BSA) including 10 ng/μL Pacific Blue™ anti-mouse TER-119 Antibody (116232, Biolegend) on ice for 30 min and washed with PBS (0.1% BSA). After washing with PBS (-), cells were measured by an attune acoustic focusing cytometer (Applied Biosystem). The data were analyzed with FlowJo (BD). PI-positive cells were excluded from the analysis. For imaging, cells were washed with PBS (-) and stained with Hoechst 33342 without trypsin detachment. After staining, the track-etched membrane was detached from the cell culture chamber, fixed on a glass slide, and imaged using a FV1000-D confocal microscopy (OLYMPUS).

### Preprocessing ELASTomics sequence data
We used the cellranger toolkit (version 6.1.2) to generate the count matrix with reference genome and transcriptomes of GRCh38-2020-A (human) and mm10-2020-A (mouse). Analysis of the count matrix was performed using Seurat version 4.0.4 package. We discarded cells with more than 10% (cancer cell lines) or 5% (mHSPCs, TIG-1 cells) mitochondrial gene expression or fewer than 2000 (cancer cell lines and TIG-1 cells) or 1000 (mHSPCs) counts of RNAs from the analysis. We further removed any cells with more than 100,000 (cancer cell lines) or 60,000 (mHSPCs and TIG-1 cells) counts to discard doublets. The counts of transcripts were normalized with a log-normal transformation and scaled with a scale factor of 100,000. The top 2000 variable genes were identified with the FindVariableFeatures function with an option of selection.method = "vst". The data set from DTD-labeled cells were integrated with non-labeled cells with functions of SelectIntegrationFeatures, FindIntegrationAnchors, and IntegrationData. Principal component analysis was carried out and neighborhood graphs were computed with the FindNeighbors function. The DTD count dataset was normalized by centered log-ratio (CLR) normalization for comparison with the gene expression dataset. Pearson's correlation coefficient between the amount of imported DTD and each gene expression were calculated by cor function. Significant differences in expression between the two groups were computed with the FindMarkers function. We applied Gene Set Enrichment Analysis (GSEA) to ordered sets of genes according to the correlation coefficient or fold change using gseGO function of clusterProfiler (ver.4.6.0). Coefficients of multiple regression between the amount of imported DTD and each gene expression were cross-validated and calculated using cv.glmnet and glmnet functions.

### Reporting summary
Further information on research design is available in the Nature Portfolio Reporting Summary linked to this article.

## Data availability
Raw and assembled sequencing data from this study have been deposited in NCBI's Sequence Read Archive (SRA) under accession code PRJNA928498 [https://www.ncbi.nlm.nih.gov/bioproject/928498] and PRJNA928499 [https://www.ncbi.nlm.nih.gov/bioproject/928499]. Electron microscopy images from this study have been deposited in Systems Science Biological Dynamics repository (SSBD:repository) under accession code ssbd-repos-000343 [https://doi.org/10.24631/ssbd.repos.2024.04.343]. Flow cytometry data from this study have been deposited in GitHub under accession code 10.5281/zenodo.10934353 [https://doi.org/10.5281/zenodo.10934352]. Source data are provided with this paper.

## Code availability
Source code and analysis scripts for DTD counts, flow cytometry, and AFM are available as Supplementary Software. Updated versions can be found in the branch of "Shiomi et al. 2023" at https://github.com/RIKEN-Microfluidics-Lab/ELASTomics.git and in Zenodo [https://doi.org/10.5281/zenodo.10934353].

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

## Acknowledgements

This work was supported by JST, CREST Grant Number JPMJCR2124, Japan, JSPS KAKENHI Grant Number JP21K18194 to H.S and JP22K15113, JP23H04724, and JP21KK0174 to A.S. The authors thank the Support Unit for Bio-Material Analysis, Research Resources Division, RIKEN Center for Brain Science for performing sample preparation of ELASTomics. We thank Reiko Takahashi, Asako Sakaue-Sawano, and Atsushi Miyawaki of RIKEN for their technical support on animal experiments.

## Author contributions

Conceptualization: A.S. and H,S.; methodology: A.S., K.D. and H.S.; investigation: A.S., K.N., T.K., A.T., T.I., M.S., K.T. and H.N.; formal analysis: A.S., T.K., and H.S.; writing original draft: A.S. T.K., K.N., and H.S. and editing: A.S. and H.S.; funding acquisition: A.S. and H.S.; supervision: H.S.

## Competing interests

The authors declare no competing interests.
