## [Peer Review File · Nature Communications]

High-throughput mechanical phenotyping and transcriptomics of single cellsREVIEWER COMMENTS

Reviewer #1 (Remarks to the Author):

The authors developed a novel method to simultaneously measure cell surface tension and gene expression. This method leveraged the CITE-seq protocol, which electrophoretically imports DNA-tagged dextran (DTD) molecules into cells and then counts imported DTD molecules using sequencing. The authors did experiments to validate and characterize the design and performed multiple experiments to study cell surface mechanics changes during differentiation and cellular senescence. This work is very interesting, but I have some comments that need to be addressed.

- 1) The authors should provide more details about how to fabricate the device so that their method can be replicated.
- 2) This work lacks both technical and biological replicates, which is difficult to evaluate how robust it is.
- 3) Cells were electroporated to deliver the DTD molecules, will this process affect (e.g., high voltage) the cell states? Also, cells are needed to be retrieved from the device before 10x experiment, will cell states change during the retrieval process?
- 4) What is the data quality of this method? How many genes or UMIs can be obtained? The authors should provide this information from all experiments they performed.
- 5) The authors didn't show the logic behind choosing the experimental parameters, for example,
 - Page 2, line 52. Why the authors tested DTD molecules with different Stokes radii? How did they choose the range of the radii?
 - Supplementary Table 3, how did the authors determine the concentrations of DTD for different experiments?
 - Page 2, line 64. how did the authors determine the voltage and frequency for different experiments?
- 6) In Supplementary Figure 2f-i, the authors should provide more details about the simulation, it is too brief. Did the authors perform the mesh dependency test? Also, what is the purpose of simulation here?
- 7) Page 3, line 79. "tenion" should be "tension".
- 8) Fig.1b, how did the authors measure the DTD imported into MCF10A cells?
- 9) Can the authors elaborate more on the "spiked 62-nucleotides long tags free of dextran"? How was it performed?
- 10) Supplementary Figure 5i, why there are a lot of signals from other clusters?
- 11) Can the authors discuss why "expression of RPL11 and RPL37 showed an inverse correlation with DTD abundance"?
- 12) Can the authors explain the following paragraph? "Interestingly, GSEA showed enrichment in myosin heavy chain class II protein complex binding (GO:0023026) and symporter activity (GO:0015293), which were insignificant for genes ordered by fold change between the two cell types (Fig. 1k, l and Supplementary Figure 5j). The results underscore the importance of cytoskeleton-related proteins that commonly play a role in the regulation of cell surface tension in both cell types." The purpose of this paragraph is not clear.
- 13) Fig. 2e, it seems that DTD signals are homogeneous. How did the authors conclude that "The projection of DTD counts on the UMAP showed that nanopore electroporation enabled import of DTD to

cells in various states”? It seems that the DTD signals cannot differentiate different states.

14) Page 7, line 241. Why the authors only discussed the RRAD gene, how it was selected?

15) Supplementary Figure 1d, the authors should provide more information about how “electrophoretic mobility of dextran and DTD measured by a microfluidic approach”?

16) Fig. 1j, what do the colors mean?

17) Fig. 3b, why different PDLs were used compared to Fig. 3a?

18) Fig. 3c-e, why PCA was used here instead of UMAP?

Reviewer #2 (Remarks to the Author):

Strategies to measure cellular mechanical phenotypes and their relations with cellular functions have been long searched for. In this manuscript, Shiomet al. reported that ELASTomics (electroporation-based lipid-bilayer assay for cell surface tension and transcriptomics) to evaluate the surface mechanical properties of the cells, where the authors introduced oligonucleotide-labelled macromolecules into cells via nanopore electroporation and quantified the number of imported macromolecules by sequencing. The results showed that high cellular surface tension positively correlates with the expression of biomarkers indicating low invasiveness in cancer cells, erythroid differentiation in HPSCs, and increased level of senescence in fibroblast cell line. The key hypothesis the authors relied critically on is that cell surface tension affects the size of the pore formed in a plasma membrane under nanopore electroporation and thus determines the introduction of macromolecules. Unfortunately, the validation of such a hypothesis is not well presented. In addition, the number of nanopores formed by electroporation on different cells (same cell line or different cell types) within a given time period is not known and the consequence brought by this difference on the number of imported macromolecules is not discussed. More importantly, would the surface size of the cell to be electroporated affect the total number of imported macromolecules. Overall, the ELASTomics design is novel and inspiring, fulfilling the goal of high throughput mechano-phenotyping, but the biological characterization, justifications, and scientific writing need major improvements.

1. The overall characterization of the cells after nanopore electroporation needs to be addressed. General data regarding the viability of target cells, calcium levels, stability of transcriptome and surface tension levels after the electroporation are missing from the manuscript. Meanwhile, the detected cells in manuscript varied from human cells to mouse cells. The detection efficiency of cells from different species were not compared and explained in detail, weakening the applicability of the whole system. The results section is quite descriptive and lack of quantitative information of the results.

2. It is not clear how the authors determine the contribution between cortical tension and plasma membrane tension in the total cell surface tension (Page 3). How do the cortical and plasma membrane tensions change in different cellular statuses as well as across cell types?

3. For the section of stroke radius of DTD and translocation, justification of analyzing DTD with different molecular weights or stroke radius is missing. Meanwhile the section should be listed along with the

characterization parts instead of after the biological results.

4. The authors stated notable cell-to-cell heterogeneity in DTD counts within and across different cell types and accounted for such heterogeneity to the different surface tension of individual cells (Line 112). The reviewer wonders if this is due to the fluctuation of the measurement by ELASTomics?

5. The authors found the permeability ratio for a DTD pair was independent of cell type and the effect of cell size can be normalized by the permeability ratio. It is not clear how this normalization can be done. Also, the cell size varies within and across the different cell types. How would this affect the number of imported DTD macromolecules?

6. In the section of cellular senescence, the choice of TIG-1 cell line rather than primary fibroblast is questionable. The senescence of cell line is different from the primary cells, lacking many functional features. Therefore, some critical genes mediating the surface mechanism may be missing from the current results.

7. The authors demonstrated that senescent TIG-1 population increased cell surface tension than did young TIG-1 population by ELASTomics. Validation of this observation by other methods such as atomic force microscopy is needed.

8. The authors identified the role of RRAD in regulating cell surface tension and functional changes during senescence of TIG-1 cells. Is it also involved in determining the different mechanical phenotypes of cancer cells the author studied? Or RRAD is specially functioned in TIG-1 cells?

9. In Fig. 1b, authors proved that Cytochalasin D-mediated loss of in-plane membrane tension in the noninvasive MCF10A cells caused a decrease of detectable signals. However, it might be because surface tension is high in MCF10A cells, more data on the influences of loss of cortical tension in low surface tension level cells need to be provided to illustrate the working detect range of cell surface tension levels. Also, the authors need to prove the influence of actin-membrane adhesion on the detectable signals, which is another essential factor influencing the cell surface tension.

10. In Fig. 1c and line 167, critical sentences describing the detailed values of detectable signals are missing. Only by indicating one group is higher than the other one is inadequate to illustrate the accurate detecting efficiency of the system. Similar issues also occur in other figures like Fig. 1d. In line 109, the authors indicated that results acquired from optical tweezers are similar to the ones from proposed system. However, the comparison between the two results and advantages over the optical tweezers is missing.

11. In line 135, The title is too brief. Only listing two subjects for analysis is inadequate to conclude the following results. Same issues occur in the following titles. In line 159, authors chose genes regarding MHC II binding and symporter activities to underscore that ELASTomics detected unique target proteins that are nonsignificant in conventional fold change of the gene expression detection. However, in Fig. S5j, similar genes regarding sodium ion transporter were neglected and not discussed, which also influence the surface tension. Meanwhile, it seems that ELASTomics failed to show significant differences

for essential genes that regulate the surface tension, including cadherin binding, cell adhesion molecule binding, actin binding and so on, which are significantly different with conventional fold change detection. Therefore, the accuracy of the ELASTomics detection on potentially meaningful marker genes is questionable.

12. In line 217, authors utilized 50V for electroporation of fibroblast, while in line 64, authors used 40V for cancerous cells, and 75V for nonadherent cells. The selection standard and optimization of the electroporation voltage for the best detection outcome is missing in the text.

13. In line 177, gene names should be normatively written as SCA1, CD48, and CD150. In line 569, equation is incomplete to show. These typos should be carefully checked.

Reviewer #3 (Remarks to the Author):

In this manuscript the authors describe a new technique called ELASTomics that can simultaneously profile cell surface mechanics information and tie it to a cell's transcriptomic signature at single cell resolution. For this study the authors have designed DNA tagged dextran (DTD) molecules of different sizes which are then electroporated into cells using track etched membrane platforms. The intracellular DTD molecule abundance can then be read out by sequencing the tagged DNA as part of the cell's transcriptome and proteome (CITE-seq). The cargo delivery quantity via nano-electroporation directly correlates to cell surface tension and thus serves as a surrogate for cell surface tension information. Hence, this technique can be used as a method to study cell surface mechanics and connect it to gene expression.

The authors first characterize the ELASTomics method by showing the positive correlation between DTD abundance and cell membrane tension by chemically modifying cytoskeletal proteins to reduce membrane tension and comparing the results to controls. They also show the same correlation in different cell types with known differences in membrane tension. Next, they use ELASTomics to quantify the transient differences in surface mechanical properties of hematopoietic stem/progenitor cells during erythroid differentiation. They find that membrane tension increases along the erythroid lineage and genes like spectrin alpha and beta that regulate erythroid membrane structure and membrane tension are positively correlated to DTD counts. Finally, they investigate the change in cell membrane tension with increasing cellular senescence using ELASTomics. They find that cell membrane tension increases with increasing senescence and genes like RRAD contribute to increasing membrane tension and senescence.

Overall, this manuscript is well written, and the experiments are well executed. Although nano-electroporation is an existing method, its combination with a reporter for cell surface tension and transcriptomics is novel and provides an interesting method for investigating structural mechanics of cells along with their gene expression at the single cell level. The experimental designs and data analysis are adequate. The inferences and conclusions are sufficiently supported by the results. The methodology is sound and described in detail which should allow reproducibility. However, the following comments need to be addressed prior to publication:

1) A brief discussion of previous literature of nano-pore electroporation is recommended in the introduction to provide context:

a) W. Kang et al., Microfluidic device for stem cell differentiation and localized electroporation of postmitotic neurons, Lab on a Chip, 2014

b) Cao et. al., Nontoxic nanopore electroporation for effective intracellular delivery of biological macromolecules, PNAS, 2019

c) Patino et. al., Multiplexed high-throughput localized electroporation workflow with deep learning-based analysis for cell engineering, Science Advances, 2022

d) Pathak et. al., Cellular Delivery of Large Functional Proteins and Protein–Nucleic Acid Constructs via Localized Electroporation, Nano Letters, 2023

e) Nathamgari et al., Localized electroporation with track-etched membranes, PNAS, 2019

2) The authors use 100 nm sized nanopores for the experiments. What pore density was used for the studies? Pore density has a significant impact on the electroporation parameters and the cargo delivery dosage. So, this value must be reported in the main text.

3) In Fig 1h, the MCF10A cluster seems to be separated by the voltage applied (0 V control vs 40 V) with the 40 V treated cell cluster showing DTD expression as expected. However, could the authors discuss why there might be this separation of the 2 groups in UMAP clustering? In this context previously it has been shown that nano-electroporation could elevate cell stress transiently –

a) Mukherjee et. al., Single cell transcriptomics reveals reduced stress response in stem cells manipulated using localized electric fields, Materials Today Bio, 2023.

Could there be some cell stress gene overexpression that is causing this clustering separation? The impact of electroporation induced stress should be discussed in context of the previous reports and the inferences drawn in this study.

4) In Fig 1j the authors correlated the DTD counts to the gene expression ratio in two cell types. Could the authors identify some key genes that show statistically significant positive correlation and discuss the biological implications? Similarly, how they have done for subsequent erythroid differentiation and cellular senescence studies.

5) For the HSPC differentiation and cellular senescence studies a pathway analysis like GSEA is recommended like was done for the cell line comparison. It may reveal additional insights into the biological processes like in the first study.

Reviewer #1 (Remarks to the Author):

The authors developed a novel method to simultaneously measure cell surface tension and gene expression. This method leveraged the CITE-seq protocol, which electrophoretically imports DNA-tagged dextran (DTD) molecules into cells and then counts imported DTD molecules using sequencing. The authors did experiments to validate and characterize the design and performed multiple experiments to study cell surface mechanics changes during differentiation and cellular senescence. This work is very interesting, but I have some comments that need to be addressed.

We appreciate the reviewer's insightful comments and constructive questions. We were encouraged by the reviewer's enthusiasm for our approach. In our revised manuscript, we have addressed all the concerns in full by adding new experiments and analyses. We respond to the questions in a point-by-point format as follows:

1) The authors should provide more details about how to fabricate the device so that their method can be replicated.

We have added extended figures and elaborated the device fabrication method to the Method section as follows:

“To fabricate the culture chamber, a hole with an 8-mm diameter was punched at the center of the PDMS slab removed from mold (Supplementary Fig. 2b). Track-etched membranes (Isopore, VCTP04700, Merck; pore size: 100 nm; pore density: $6.04 \pm 0.52 \mu\text{m}^{-2}$) were cut to the same size as the bottom of the PDMS slab and glued on the bottom of the PDMS slab using the uncured PDMS (Supplementary Fig. 2c). To fix the track-etched membrane, the assembled PDMS slabs were cured in an oven at 70°C for 30 min. The cell culture chambers were sterilized by a UV exposure for more than 10 min in a safety cabinet before use (Supplementary Fig. 2d).

To fabricate the electrode holder, the electrode holder was directly printed using the 3D printer with the same procedure as described above (Supplementary Fig. 2e). A platinum electrode (PT-356320, Nilaco) was glued on the bottom of the electrode holder with power cable using the pre-cured PDMS and fixed in an oven at 70°C for 30 min.”

2) This work lacks both technical and biological replicates, which is difficult to evaluate how robust it is.

We have intensively assessed the robustness of our approach by creating technical and biological replicates. In the revised manuscript, we have included the assessment to show the robustness of ELASTomics.

To effectively investigate the reproducibility of importing molecules via nanopore electroporation, we carried out nanopore-electroporation of cells using FITC-BSA as a substitute of DTD and read out the quantity of imported molecules per cell by a flow cytometry (Supplementary Fig. 3a, b). We have included the additional experimental data that shows the reproducibility such as HeLa cells shown in Supplementary Fig. 3a, b, and 5e, f.

Additionally, we have also assessed the reproducibility of ELASTomics by running two independent experiments with MCF10A cells at 40 V of the applied voltage (Fig. 1b, c). The data from the two independent experimental runs merged in the UMAP, supporting the reproducibility of ELASTomics (Supplementary Fig. 3s-u).

To discuss the robustness of ELASTomics, we have added the following texts in the Result section:

“The quantity of the imported molecules was reproducible under the same condition (Supplementary Fig. 3a, b, and Supplementary Fig. 5e, f).”

“To further minimize the effect of the nanopore-electroporation and the DTD import on the gene expression analysis, we collected single-cell data with non-nanopore electroporated cells and integrated with the ELASTomics data before analysis (Fig. 1b, c and Supplementary Fig. 3s-u). This process allowed us to robustly analyse the gene expression.”

3) Cells were electroporated to deliver the DTD molecules, will this process affect (e.g., high voltage) the cell states? Also, cells are needed to be retrieved from the device before 10x experiment, will cell states change during the retrieval process?

As the reviewer pointed out, a high voltage affects the cell states. To minimize the unwanted perturbation due to the nanopore-electroporation and importing DTD into cells, we optimized the protocol by tuning the magnitude of the applied voltage and the retrieval process. As for the applied voltage, we assessed the cell viability at various applied voltages by staining the cells with propidium iodide and measuring them by flow cytometry to determine the applied voltage that yielded more than 90% of cell viability (Supplementary Fig. 3a, b).

As for the nanopore-electroporation process, we optimized the buffer as calcium free solution to avoid cell death and changes in gene expression due to calcium influx (Supplementary Fig. 2f).

We have added the information on the buffer consideration as follows:

“To prevent cell stimulation and cell death due to calcium influx, we selected calcium-free solutions, PBS (-) and HEPES based buffer (20 mM HEPES/NaOH, pH 7.0 and 260 mM sucrose), for nanopore electroporation.”

As for the retrieval process, we optimized the duration of incubation as 2 min after applying nanopore electroporation and before replacing the culture medium to reseal the pores in the phospholipid bilayers before the retrieval. The 2-min incubation was critical to keep the cell viability and prevent the calcium stimulation. After the retrieval of the cells by trypsinization, we suspended cells in a culture medium and kept on ice, and then carried out the scRNA-seq within 1 h.

We have added this information in retrieval process as follows:

“We also note that incubation for 2 minutes after applying the electrical pulses was critical to reseal the pores in the phospholipid bilayers and to protect the cells from calcium stimulation and cell death. After trypsinization, cells were kept on ice to suppress changes in gene expression and performed scRNA-seq within 1h.”

To further minimize the effect of the nanopore-electroporation and the DTD import on the gene expression analysis, we collected single-cell data with non-nanopore electroporated cells and integrated with the ELASTomics data before further analysis. The optimized pipeline allowed to successfully integrate nanopore-electroporated cells with control cells in UMAP as shown in Supplementary Fig. 3s-u.

We have added the information about the minimize the effect of the nanopore-electroporation in new result section (see our response to #2).

To highlight the importance of the optimization of the applied voltage, we performed ELASTomics with PC-3 cells treated with 75 V pulses. The resulted gene expression showed an

increased number of variable genes between PC-3 cells nanopore-electroporated at 75 V and the control cells (Supplementary Fig. 3d-g).

We have added the information about the effect of applied voltage in new result section (see the response to #5).

4) What is the data quality of this method? How many genes or UMIs can be obtained? The authors should provide this information from all experiments they performed.

We have added the information about the detected genes and UMI in each Result section as follows:

•We identified 3,804 PC-3 cells (median of 10,805 UMI and 3,432 genes detected per cell), 2,275 MDA-MB-231 cells (median of 4,594 UMI and 1,924 genes detected per cell), 1,575 MCF7 cells (median of 13,308 UMI and 3,536 genes detected per cell), and 2,083 MCF10A cells (median of 6,124 UMI and 2,052 genes detected per cell). We also identified 4,018 PC-3 cells (median of 10,800 UMI and 3,440 genes detected per cell), 1,691 MDA-MB-231 cells (median of 6,613 UMI and 2,494 genes detected per cell), 3,087 MCF7 cells (median of 13,136 UMI and 3,363 genes detected per cell), and 3,412 MCF10A cells (median of 4,306 UMI and 1,531 genes detected per cell) as the controls.

•We identified 10,011 mHSPCs with a median of 6,893 UMI and 1,919 genes detected per cell.

•We identified 4,654 TIG-1 cells (applied voltage: 0 V) with a median of 13,130 UMI and 3,648 genes and 4,711 TIG-1 cells (applied voltage: 50 V) with a median of 19,207 UMI and 4,474 genes detected per cell.

5) The authors didn't show the logic behind choosing the experimental parameters, for example, - Page 2, line 52. Why the authors tested DTD molecules with different Stokes radii? How did they choose the range of the radii?

The critical radius of the reversible pores formed in a lipid bilayer is theoretically estimated as 15 nm (Mukherjee, P., et al., *ACS Nano*, 12:12118, 2018). We thus designed the DTD that has similar or smaller radii than 15 nm. We also aimed to probe the radius of pores by profiling the difference in the imported quantity due to the radii of DTD molecules.

We have added the information in Result section as follows:

*“We designed the DTD so that the Stokes radii (4.1 ± 0.0 nm– 17.0 ± 12.2 nm) cover the critical radius, 15 nm, which is the theoretical maximum radius to reseal at equilibrium (Mukherjee, P., et al., *ACS Nano*, 12:12118, 2018).”*

- Supplementary Table 3, how did the authors determine the concentrations of DTD for different experiments?

DTD concentration is an important parameter that influences the number of DTD counts: a higher concentration yields more DTD counts. Prior to performing ELASTomics, we applied nanopore-electroporation to cells with various concentrations of DTDs and quantified the yielded library by qPCR to estimate the counts per cell. To gain the maximum signal-to-noise ratio, we used the maximum concentration that we could prepare, depending on the type of DTD.

We have added the optimization of DTD concentrations in Method section as follows:

DTD concentration is an important parameter that influences the number of DTD counts: a higher concentration yields more DTD counts. To gain the maximum signal-to-noise ratio, we here used the maximum concentration that we could prepare, depending on the type of DTD as summarised in Supplementary Table 3.

- Page 2, line 64. how did the authors determine the voltage and frequency for different experiments?

In our preliminary experiments, we explored various conditions of applied voltages. For instance, in the case of HeLa cells, we observed the 500 pulses (width of 5 ms) at 20 Hz resulted in the highest amount of imported FITC-BSA without compromising the cell viability (Supplementary Fig. 3a, b). We also found that the higher applied voltage increased the amount of imported molecules via nanopore electroporation, improving the sensitivity of the measurement of the cell surface tension (Supplementary Fig. 3b, c). However, as we discussed in the response to #3, the excessive magnitude of the applied voltage gave negative impact on the cell viability and distorted the gene expression (Supplementary Fig. 3d-g). To balance between the sensitivity and the negative impact on the cells, we assessed the quantity of imported molecules and cellular viability by performing the nanopore electroporation with FITC-BSA followed by flow cytometry (Supplementary Fig. 3h-m). We determined the conditions of ELASTomics with >90% survival as 40 V for the four cancer cell lines (PC-3, MDA-MB-231, MCF7, and MCF10A cells), 75 V for mHSPCs and 50 V for TIG-1 cells.

We added the new section about the experimental condition in Result section as follows:

“Optimization of nanopore electroporation for ELASTomics

To balance between the sensitivity and the negative impact of nanopore electroporation on the cells, we assessed the quantity of imported molecules and cellular viability by performing the nanopore electroporation with fluorescein isothiocyanate-labelled bovine serum albumin (FITC-BSA; Stokes radius of 3.6 ± 1.4 nm) as a substitute of DTD at various voltage conditions followed by flow cytometry (Supplementary Fig. 3a, b). The amount of imported FITC-BSA correlated well with that of DTDs (Supplementary Fig. 4a-d).”

“Our experimental data showed that the higher applied voltage increased the amount of imported molecules, improving the sensitivity of the measurement of the cell surface tension (Supplementary Fig. 3c). However, the excessive magnitude of the applied voltage gave negative impact on the cell viability and distorted the gene expression (Supplementary Fig. 3d-g). On the basis of the experimental surveillance, we determined the conditions of ELASTomics with >90%

viability as 40 V for the four cancer cell lines (PC-3, MDA-MB-231, MCF7, and MCF10A cells), 75 V for mouse haematopoietic stem/progenitor cells (mHSPCs) and 50 V for TIG-1 cells (Supplementary Fig. 3h-m). The quantity of the imported molecules was reproducible under the same condition (Supplementary Fig. 3a, b, and Supplementary Fig. 5e, f). We confirmed that nanopore electroporation can be applied to various cell lines including HeLa, PC-3, MDA-MB-231, MCF7, MCF10A, TIG-1, OVCAR-3 (adherent cells, human), CHO-K1 (adherent cells, Chinese hamster), GEM-81 cells (adherent cells, goldfish), primary mHSPCs (suspension cells, mouse), and K562 (suspension cells, human) (Supplementary Fig. 3h-r). We note that non-adherent cells such as K562 and mHSPC require higher applied voltages than adherent cells to import a similar quantity of molecules by nanopore electroporation.”

6) In Supplementary Figure 2f-i, the authors should provide more details about the simulation, it is too brief. Did the authors perform the mesh dependency test? Also, what is the purpose of simulation here?

We note that the original text was a bit misleading when it refers to Supplementary Fig. 2f-i. Supplementary Fig. 2f and g were experimentally measured voltage and current, respectively. Supplementary Fig. 2h and i were the results from the numerical simulations.

We have reevaluated the conditions for the numerical simulations and used appropriate parameters to recapitulate the experimental conditions. Further, as the reviewer recommended, to validate the numerical result, we additionally performed a mesh dependency (Supplementary Fig. 2j-l).

To clearly describe the purpose of the numerical simulation and details, we have revised the texts in Supplementary Methods as follows:

“To visualize the focused electric field in the vicinity of the nanopore, we simulated the electric field in our experimental system using COMSOL Multiphysics finite-element-analysis software (COMSOL Inc) along with AC/DC Module (steady state). We assumed each nanopore is an independent system and identical to the others, and thus we modelled the single nanopore as an axisymmetric cylinder with 25-μm height and 100 nm in diameter filled with the HEPES based buffer. The PBS in a culture chamber (on top of the track-etched membrane, 0.633 Ωm) and the

HEPES based buffer in the electrode holder (under the track-etched membrane, 44.8 Ω m) were both simplified as cylindrical media with 1-mm height and 0.41- μ m diameter, which corresponded to the average distance between nanopores. We analytically calculated the voltage difference between the upper and lower boundaries of simulated area. Assuming that the top and bottom electrodes (in the arrangement shown in Supplementary Fig. 2f) were subjected to voltages of 40 V and 0 V, respectively, the voltages at the upper and lower boundary of the simulated area were estimated as 38.6 V and 27.5 V, respectively. We thus applied a voltage difference of 11.1 V between the top and the bottom boundaries. Boundary surfaces other than the top and bottom were assumed to be insulators. We build the mesh through a physics-controlled mesh sequence type with the parameters of 'Extremely fine' element size, 0.1- μ m of maximum element size, and 0.0205- μ m of minimum element size. The numerical result showed the focused electric field rapidly decays as the distance from the nanopore within the order of 1 μ m."

7) Page 3, line 79. "tenion" should be "tension".

Thanks for pointing out our mistake. We have changed the section title.

8) Fig.1b, how did the authors measure the DTD imported into MCF10A cells?

We counted the number of DNA tags derived from DTD imported into MCF10A cells with the sequencing data. To clarify it, we have written the text as:

"Following cytochalasin D treatment or vehicle on MCF10A cells, we performed ELASTomics with the respective MCF10A cells, where we prepared scrNA-seq libraries and counted the number of DNA tags derived from DTD in single cells (Fig. 1b)."

Line 93 in the original manuscript, "control cells treated with the vehicle" was slightly unclear. We thus rephrase it as "cells treated with vehicle" and the label "Control" in Fig. 1b was changed as "Vehicle".

9) Can the authors elaborate more on the "spiked 62-nucleotides long tags free of dextran"? How was it performed?

We simply mixed the cell-hashtag into DTD solution for nanopore-electroporation. The information of the cell-hashtags was missing in the method of the original manuscript. We have added the information as follows:

"The cell culture chamber was then placed on the electrode holder prefilled with 150 μ L of a HEPES based buffer including DTD and cell-hashtags at the concentration summarised in the Supplementary Table 3."

	4 kDa DTD (μM)	10 kDa DTD (μM)	40 kDa DTD (μM)	70 kDa DTD (μM)	150 kDa DTD (μM)	500 kDa DTD (μM)	Cell-hashtag (μM)
TIG-1 cells	0.33	1.00	1.50	0.50	1.50	0.50	-
Mouse haematopoietic stem/progenitor cells	1.00	1.00	1.00	0.50	1.00	0.50	-
PC-3, MDA-MB-231, MCF7, MCF10A cells	1.00	1.00	1.00	0.50	1.00	0.50	1.00

10) Supplementary Figure 5i, why there are a lot of signals from other clusters?

To obtain the data in Supplementary Fig. 5i, we performed scRNA-seq with pooled cells including PC-3 and MCF7 cells labelled with the respective cell-hashtags via nanopore electroporation. PC-3 appeared to be low cell surface tension that led to relatively low counts of cell-hashtags and thus the detection of their cell-hashtags was susceptible to the background noise. However, this noise does not detract from the identification of cell type and batch of clusters, because we identified the cell types based on the gene expression.

We have added this note on the background noise in the figure legend of the Supplementary Fig. 5i (Supplementary Fig. 7i as revised figure) as follows:

“The hashing tag for PC-3 was susceptible to the background noise in j because of the low counts due to the low surface tension of PC-3 cells. However, the low specificity of the cell hashing tag does not detract from the identification of cell type and batch of clusters, because we identified the cell types based on the gene expression.”

11) Can the authors discuss why “expression of RPL11 and RPL37 showed an inverse correlation with DTD abundance”?

The dysfunction of p53, a tumour suppressor, occurs in most human malignancies. As previously reported, ribosomal protein L37 (RPL37) regulates p53 levels by binding to Mdm2, a RING-type E3 ubiquitin ligase (Llanos, S., & Serrano, M., *Cell cycle*, 9:4005, 2010; Daftuar, L., et al., *PloS one*, 8:e68667, 2013). The ribosomal protein L11 (RPL11) also binds to Mdm2 to inhibit the ubiquitination and degradation of p53 (Zhang Y., et al., *Mol. Cell Biol.*, 23: 8902, 2003). Therefore, RPL11 and RPL37 are potentially involved in the low cell surface tension coupled with cancer malignancy via p53 axis.

We added them in Discussion section as follows:

*“ELASTomics data on MCF7 and MDA-MB-231 cells identified the shared negative correlation of RPL11 and RPL37 expressions with DTD counts (Fig. 1j). RPL37 regulates expression level of p53, which is a tumour suppresser, by binding to Mdm2, a RING-type E3 ubiquitin ligase (Llanos, S., & Serrano, M., *Cell cycle*, 9:4005, 2010; Daftuar, L., et al., *PloS one*, 8 :e68667, 2013). RPL11 also binds to Mdm2 to inhibit the ubiquitination and degradation of p53 (Zhang Y., et al., *Mol. Cell Biol.*, 23: 8902, 2003). Taken together, the negative correlation of RPL11 and RPL37 with*

DTD counts may imply that the regulation of the cell surface tension via p53 axis is conserved among MCF7 and MDA-MB-231 independent of the cancer cell type.”

12) Can the authors explain the following paragraph? “Interestingly, GSEA showed enrichment in myosin heavy chain class II protein complex binding (GO:0023026) and symporter activity (GO:0015293), which were insignificant for genes ordered by fold change between the two cell types (Fig. 1k, l and Supplementary Figure 5j). The results underscore the importance of cytoskeleton-related proteins that commonly play a role in the regulation of cell surface tension in both cell types.” The purpose of this paragraph is not clear.

We here wanted to highlight the enriched functions that were solely detected by the measurement of the cell surface tension rather than just difference due to cell types and shared among MCF7 and MD-MB-231 cells. To find such functional enrichment coupled with the cell surface tension, we compared GSEA performed by a gene set ordered by the correlation against a standard GSEA performed by gene set ordered by the fold change among MCF7 and MDA-MB-231 cells. To explain this clearly, we have rewritten the sentences in the Result section as follows:

“Interestingly, GSEA based on the correlation with the DTD counts showed enrichment in some gene sets such as myosin heavy chain class II protein complex binding (GO:0023026) and symporter activity (GO:0015293), which were insignificant in a standard GSEA based on the fold change of gene expressions between the two cell types (Fig. 1k, l and Supplementary Fig. 8c), highlighting the shared enrichment in the cytoskeleton-related proteins among MCF7 and MDA-MB-231 cells.”

13) Fig. 2e, it seems that DTD signals are homogeneous. How did the authors conclude that “The projection of DTD counts on the UMAP showed that nanopore electroporation enabled import of DTD to cells in various states”? It seems that the DTD signals cannot differentiate different states.

Because ELASTomics needs to settle the cells on the track-etched membrane to expose the intense electric field to the cell membrane, the non-adherent and primary cells are challenging. We here do not claim that the DTD signal can clearly differentiate all the different cell states. But we claim that our optimised protocol for non-adherent cells enabled to nanopore-electroporate and deliver DTD to the primary and non-adherent cells. As the reviewer pointed out, the DTD signals seems relatively homogeneous and slightly difficult to see the difference due to the cell states. We thus highlight the cell type dependence of the DTD counts in Fig. 2f.

To emphasise the technical challenge of ELASTomics applying to non-adherent cells, we have rewritten in Result section as follows:

“To apply ELASTomics to primary non-adherent cells, which are challenging cell types, we allowed cells to settle on the track-etched membrane precoated with 100 $\mu\text{g}/\text{mL}$ fibronectin for 1 h and then applied 75 V for nanopore electroporation.”

“The projection of DTD counts on the UMAP showed that nanopore electroporation enabled import of DTD to the non-adherent cells in various states (Fig.2e).”

14) Page 7, line 241. Why the authors only discussed the RRAD gene, how it was selected?

ELASTomics identified 210 genes involved in the increase in cell surface tension along cellular senescence in TIG-1 cells. Of 210 genes, *RRAD* was the fourth candidate following two long non-coding RNAs (*AC007952.4*, *AC091271.1*) and *KLF2* based on the magnitude of the coefficient in the multiple-regression analysis. Additionally, we performed bulk RNA-seq for TIG-1 cells at different PDLs to analyse these genes and we have confirmed that *RRAD* was significantly upregulated with increasing PDLs, while not *KLF2* (Fig. 3h, i, and Supplementary Fig. 11e-m).

We have added the reasoning to select *RRAD* in Result section as follows:

“in which *RRAD* showed the fourth highest effect on cell surface tension following two long-non-coding RNAs, *AC007952.4* and *AC091271.1*, and *KLF2* genes (Fig. 3g).”

“In line with the findings of ELASTomics, the population of TIG-1 cells increased both the amount of imported FITC-BSA and the expression of *RRAD* as PDL increased while not *KLF2*, indicating that cell surface tension and *RRAD* expression are related during cellular senescence.”

RRAD inhibits the translocation of glucose transporter-1 (GLUT1) to plasma membrane to inhibit the glucose uptake. To further confirm the importance of *RRAD* on the cell surface tension via glycolysis pathway, we additionally have carried out pseudo-inhibition of glucose uptake using 2-deoxy-D-glucose that increased cell surface tension (Added in Method and Fig. 3l).

We added them in Result section as follows:

“*RRAD* represses glycolysis mainly through inhibition of glucose transporter-1 (GLUT1) translocation to the plasma membrane (Zhang, C. et al., *Oncotarget*, 5:5535, 2014).”

“In addition, we have confirmed that inhibition of glycolysis with 2-deoxy-D-glucose (2-DG) increases the cell surface tension of TIG-1 cells. (Fig. 3l). The results indicate that *RRAD* contributes to an increase in cell surface tension during senescence via glycolysis in TIG-1 cells.”

15) Supplementary Figure 1d, the authors should provide more information about how “electrophoretic mobility of dextran and DTD measured by a microfluidic approach”?

We have included the detailed method to analyse the electrophoretic mobility of dextran and DTD in method section as follows:

“We used a previously reported microfluidic approach (Milanova, D., et al., 32:3286, 2011) to measure electrophoretic mobilities. We used HEPES based buffer as the running buffer consisting of 20 mM HEPES and 260 mM sucrose (pH 7.0, resistivity 35.1 Ω m). We mixed each DTDs or dextrans and rhodamine B in the running buffer and introduced it into a cross-patterned microchannel (NS12A, Caliper Life Sciences) filled with the running buffer. We applied voltage to the channel via a computer-controlled voltage source (HVS448-3000D-LC, Labsmith), leading to the migration and separation of the analytes and rhodamine B in a microchannel branch (45.59 mm from cross junction to outlet) during current measurement (Keithley 6487 Picoammeter Voltage Source, Keithley Instruments). We observed the migration via epi-microscopy at 5 mm (20 mm for 150-kDa DTD) downstream from the cross junction. We fit the fluorescence intensities by Gaussian models using the ‘fit’ function in MATLAB and estimate the migration time of the analytes ($t_{migration, analyte}$) and rhodamine B ($t_{migration, RB}$) from the peak. We calculated the electrophoretic mobility of the analyte ($\mu_{analyte}$) from the equation below.

$$\mu_{analyte} = \frac{AL}{\rho I} \left(\frac{1}{t_{migration, analyte}} - \frac{1}{t_{migration, RB}} \right)$$

Here, A is the cross-sectional area of the microchannel, I is the current of the separation channel, L is the distance from the cross junction to the observation point, and ρ is the buffer resistivity.”

16) Fig. 1j, what do the colors mean?

We apologise that we did not explain the red points in the original manuscript. Red points in Fig. 1j show the genes with the greater log fold-change than 2.5 or less than -2.5. We have added the information about that in the figure legend as follows:

“Red points are genes with the log fold-change in the average expression greater than 2.5 or less than -2.5.”

17) Fig. 3b, why different PDLs were used compared to Fig. 3a?

We have additionally carried out the experiments with TIG-1 cells as the same PDL number used in ELASTomics and have revised the figure.

18) Fig. 3c-e, why PCA was used here instead of UMAP?

In the original manuscript, we thought the trajectory of the senescence would be clearer in PCA than UMAP. In the revised manuscript, we have changed it from PCA to UMAP in Fig. 3c-e.

Reviewer #2 (Remarks to the Author):

Strategies to measure cellular mechanical phenotypes and their relations with cellular functions have been long searched for. In this manuscript, Shiomi et al. reported that ELASTomics (electroporation-based lipid-bilayer assay for cell surface tension and transcriptomics) to evaluate the surface mechanical properties of the cells, where the authors introduced oligonucleotide-labelled macromolecules into cells via nanopore electroporation and quantified the number of imported macromolecules by sequencing. The results showed that high cellular surface tension positively correlates with the expression of biomarkers indicating low invasiveness in cancer cells, erythroid differentiation in HPSCs, and increased level of senescence in fibroblast cell line. The key hypothesis the authors relied critically on is that cell surface tension affects the size of the pore formed in a plasma membrane under nanopore electroporation and thus determines the introduction of macromolecules. Unfortunately, the validation of such a hypothesis is not well presented. In addition, the number of nanopores formed by electroporation on different cells (same cell line or different cell types) within a given time period is not known and the consequence brought by this difference on the number of imported macromolecules is not discussed. More importantly, would the surface size of the cell to be electroporated affect the total number of imported macromolecules. Overall, the ELASTomics design is novel and inspiring, fulfilling the goal of high throughput mechano-phenotyping, but the biological characterization, justifications, and scientific writing need major improvements.

We appreciate the reviewer's insightful comments and constructive questions. We were also encouraged by the reviewer's enthusiasm for our approach. In our revised manuscript, we have addressed all the concerns in full by adding new experiments and analyses. We respond to the questions in a point-by-point response format as follows:

1. The overall characterization of the cells after nanopore electroporation needs to be addressed. General data regarding the viability of target cells, calcium levels, stability of transcriptome and surface tension levels after the electroporation are missing from the manuscript. Meanwhile, the detected cells in manuscript varied from human cells to mouse cells. The detection efficiency of cells from different species were not compared and explained in detail, weakening the applicability of the whole system. The results section is quite descriptive and lack of quantitative information of the results.

We have intensively investigated the cell viability after the nanopore-electroporation using FITC-BSA as a substitute for DTDs and propidium iodide (PI).

To minimize the impact of the nanopore electroporation and DTD import on the cell states, we optimised the magnitude of applied voltage, number of pulses and the buffer (Supplementary Fig. 3a-c).

As for the applied voltages, we have explored the magnitude, width, number and frequency of the pulses. To balance between the sensitivity and the negative impact on the cells, we assessed the quantity of imported molecules and cellular viability by performing the nanopore electroporation with FITC-BSA followed by flow cytometry and then determined the conditions of ELASTomics with >90% survival as 40 V for the four cancer cell lines (PC-3, MDA-MB-231, MCF7, and MCF10A cells), 75 V for mHSPCs, and 50 V for TIG-1 cells (Supplementary Fig. 3h-m).

As we discussed in the response #5 of Reviewer#1's comment, we added a new section about the determination of experimental condition in result section.

To eliminate the calcium influx due to the nanopore-electroporation, we have used calcium-free PBS and HEPES based buffer (20 mM HEPES/NaOH {pH 7.0} and 260 mM sucrose).

Additionally, to minimise the change in the gene expression after the nanopore-electroporation, we stored cells on ice and carried out the scRNA-seq within 1 h after the nanopore-electroporation. To further minimise the influence from the perturbation due to nanopore-electroporation, we integrated our ELASTomics dataset with non-nanopore-electroporated cells before the analysis. As we discussed in the response #3 of Reviewer#1's comment, we added the information in the result section.

In ELASTomics, we opt to measure the cell surface tension at the time point of the nanopore electroporation. We observed relatively high correlation between the quantity of imported FITC-BSA and the surface tension measured by AFM (Fig. 1d), which was done after the nanopore electroporation within 0.5-3 h, supporting the stability of the cell surface tension after the nanopore electroporation. As we performed scRNA-seq within 1 h after the nanopore electroporation, our protocol was robust enough to capture the cell surface tension and the gene expression. To address this point, we have added surface tension levels after the electroporation in the Result section as follows:

“Although the measurements of surface tension using AFM were performed between 0.5-3 h after nanopore electroporation, we observed correlation between the area-normalized quantity of imported FITC-BSA and the surface tension measured by AFM (Fig. 1d, and Supplementary Fig. 4h, i), indicating that the variation in the number of imported molecules reflected cell-to-cell variation in surface tension.”

Our assessment showed that the quantity of the imported molecules by nanopore electroporation was dependent on the adhesion mode of the cell types rather than the species. We have carried

out different cell lines including HeLa, PC-3, MDA-MB-231, MCF7, MCF10A, OVCAR-3 (adherent cells, human), CHO-K1 (adherent cells, Chinese hamster), GEM-81 cells (adherent cells, fish), and K562 (suspension cells, human) (Supplementary Fig. 3h-r). We found that our approach was robust to cells from various species but it was rather dependent on the adherent or suspension. Specifically, the non-adherent cells such as K562 cells and mouse hematopoietic

progenitor and stem cells need higher applied voltages than adherent cells to import similar quantity of molecules by nanopore electroporation.

To elaborate the point, we have rewritten the Result section as follows:

“We confirmed that nanopore electroporation can be applied to various cell lines including HeLa, PC-3, MDA-MB-231, MCF7, MCF10A, TIG-1, OVCAR-3 (adherent cells, human), CHO-K1 (adherent cells, Chinese hamster), GEM-81 cells (adherent cells, goldfish), primary mHSPCs (suspension cells, mouse), and K562 (suspension cells, human) (Supplementary Fig. 3h-r). We note that non-adherent cells such as K562 and mHSPCs require higher applied voltages than adherent cells to import a similar quantity of molecules by nanopore electroporation.”

2. It is not clear how the authors determine the contribution between cortical tension and plasma membrane tension in the total cell surface tension (Page 3). How do the cortical and plasma membrane tensions change in different cellular statuses as well as across cell types?

We apologize that the original manuscript was slightly misleading in the explanation of the contributions from the cortical tension and plasma membrane tension. ELASTomics is incapable to differentiate the contributions from those two but characterise the cell surface mechanics by abstracting it as cell surface tension, which is influenced by both of the tensions (Clark, A.G., et al, *Current Biology*, 24:484, 2014).

Instead, we here wanted to demonstrate that ELASTomics can profiles the difference in the cell surface tension due to perturbation to the actin cytoskeleton, which changes the cortical tension, as well as due to difference in the plasma membrane tension of various breast cancer cell lines.

To clarify this point and avoid confusion, we combined the two sections into one. To further validate the rational of the ELASTomics we have also added some experimental data that

demonstrate probing the change in the cell surface tension due to the perturbation in the cortical tension and plasma membrane tension (see the response to #9).

3. For the section of stroke radius of DTD and translocation, justification of analyzing DTD with different molecular weights or stroke radius is missing. Meanwhile the section should be listed along with the characterization parts instead of after the biological results.

We agree with the reviewer that we should place the effect of Stokes radius of DTD on its translocation before the biological results.

Reversible pores formed by electroporation theoretically have radii smaller than 15 nm of critical radius (Mukherjee, P., et al., *ACS Nano*, 12:12118, 2018). To probe the pore diameters in the lipid bilayer via the quantity of translocated DTD, we designed DTD with the range of Stokes radii that cover the critical radius.

Theoretically, to minimize the effect of the adhesion area on the quantity of the imported DTD, DTD counts are ideally normalized by the adhesion area of cells, because the total quantity of the imported DTD is predicted to be dependent on the adhesion area. However, we found that the surface tension measured by AFM well correlated to the total quantity of the imported molecules via nanopore electroporation (Supplementary Fig. 4h). We attribute this correlation to the correlation between the adhesion area and the cell surface tension (Xie, K., Yang, Y., & Jiang, H., *Biophysical journal*, 114:675, 2018), which was also observed in our experimental data (Supplementary Fig. 4i). We also found that the permeability ratio was susceptible to noise than DTD count. We thus primary used normalized 4kDa DTD counts using centred log ratio transformation.

4. The authors stated notable cell-to-cell heterogeneity in DTD counts within and across different cell types and accounted for such heterogeneity to the different surface tension of individual cells (Line 112). The reviewer wonders if this is due to the fluctuation of the measurement by ELASTomics?

We have respectively discussed the difference in the cell surface tension across different cell types in the “**Cell surface tension—membrane tension and plasma membrane tension**” and cell-to-cell heterogeneity within the same cell type in the “**Heterogeneity of cell surface tension**”.

At the line 112 in the original manuscript, we claim the detection of the heterogeneity in the cell surface tension within the same cell type, but not difference across the different cell types. We validated the detection of cell-to-cell heterogeneity within the same cell type by comparing the

quantity of imported molecules and the surface tension measured by AFM (Fig.1d and Supplementary Fig. 4h, i).

To further confirm the detection of the heterogeneity within the same cell type, we performed ELASTomics with TIG-1 cells at a different PDL, which were the same cell types but just at the different numbers of passages, and successfully captured the change in the cell surface tension due to the replicative senescence. Further, as the reviewer suggested, we performed the measurement of the surface tension by AFM to validate the finding by the ELASTomics (see the response to #7) and confirmed the consistent results, encouraging us to claim the detection of the heterogeneity within the same cell type.

5. The authors found the permeability ratio for a DTD pair was independent of cell type and the effect of cell size can be normalized by the permeability ratio. It is not clear how this normalization can be done. Also, the cell size varies within and across the different cell types. How would this affect the number of imported DTD macromolecules?

We agree that the total quantity of the imported DTD is theoretically predicted to be dependent on the adhesion area of the cells. However, we found that the total quantity of the imported molecule shows similar correlation with the surface tension ($r = 0.685$, see Supplementary Fig. 4h) to

the normalized quantity by the adhesion area ($r = 0.648$, see Fig. 1d). We attribute the comparable correlation to the correlation between the cell surface tension and the adhesion area ($r = 0.495$), which was reported in elsewhere (Xie, K., Yang, Y., & Jiang, H., *Biophysical journal*, 114:675, 2018) and was also observed in our experimental data (Supplementary Fig. 4i). On the basis of this experimental result, we primary used normalized 4kDa DTD counts using centred log ratio transformation to correlate the cell surface tension in our analysis.

Although we do not intensively leverage the different sized DTDs, we have explored the permeability ratio in Supplementary Fig. 6 to show the usefulness. We found that the permeability ratio (the slope of the data points) was independent of the cell types, encouraging us to extend our findings to other cell types.

To discuss the effect of cell size, we have added the information in the Result section as follows: “Interestingly, the total quantity of imported molecules into a cell by nanopore electroporation showed comparable correlation with the surface tension (Supplementary Fig.4h), although it is

predicted to be proportional to the adhesion area, i.e., the number of nanopores in the track-etched membrane beneath the cell membrane. We attribute this to the dependence of the adhesion area on the cell surface tension (Xie, K., Yang, Y., & Jiang, H., Biophysical journal, 114:675, 2018), which was also observed in our data (Supplementary Fig. 4i) supporting that the effect of the adhesion area on nanopore-electroporation reinforces, rather than weakens, the correlation between surface tension and the amount imported molecules. "

6. In the section of cellular senescence, the choice of TIG-1 cell line rather than primary fibroblast is questionable. The senescence of cell line is different from the primary cells, lacking many functional features. Therefore, some critical genes mediating the surface mechanism may be missing from the current results.

We agree with the reviewer's opinion that the primary fibroblast would be very interesting cells to investigate with ELASTomics.

TIG-1 cells are well established cell line to study the senescence pathway induced by replicative amplification, DNA-damage, and drug-treatment (Fujita, Y., et al., *Experimental Gerontology*, 165: 111866, 2022; Udono, M., et al., *Scientific reports*, 5:17342, 2015; Kim, G., et al., *Aging cell*, 11:617, 2012). Despite the correlation between the proliferation limit of a cell line and the life span of a species (Röhme, D., *PNAS*, 78:5009, 1981), cultured primary human fibroblasts, including TIG-1 cells, appear to behave differently from primary cells, particularly in the capacity of cell division (Sherr, C. J., and Ronald A. D., *Cell*, 102:407, 2000).

To discuss the limitation of using TIG-1 cells, we have added the information in the discussion section as follows:

"We also note that although TIG-1 cells are a well-established cell line for studying replicative senescence (Fujita, Y., et al., Experimental Gerontology, 165: 111866, 2022; Udono, M., et al., Scientific reports, 5:17342, 2015; Kim, G., et al., Aging cell, 11:617, 2012), the cultured senescent cells including TIG-1 cells behave somewhat differently than senescent cells in vivo (Sherr, C. J., and Ronald A. D., Cell, 102:407, 2000). We envision that ELASTomics would be applicable to the primary fibroblasts by integrating with rapid cell isolation protocols (Soteriou, D., et al., Nature Biomedical Engineering 7:1392, 2023; Kotaro H., et al., Life Science Alliance, 6:e202201783, 2022)."

7. The authors demonstrated that senescent TIG-1 population increased cell surface tension than did young TIG-1 population by ELASTomics. Validation of this observation by other methods such as atomic force microscopy is needed.

To validate the finding that TIG-1 increases the cell surface tension along the senescence, we have carried out the measurement of cell surface tension using AFM on TIG-1 cells at different PDLs (Supplementary Fig. 11d).

We have rewritten in the Result section as follows:

“We also confirmed that on average the surface tension of senescent TIG-1 cells was higher than that of young TIG-1 cells by AFM (Supplementary Fig. 11d).”

8. The authors identified the role of RRAD in regulating cell surface tension and functional changes during senescence of TIG-1 cells. Is it also involved in determining the different mechanical phenotypes of cancer cells the author studied? Or RRAD is specially functioned in TIG-1 cells?

We appreciate the very interesting insight by the reviewer. Although we additionally analysed the data of the cancer cells to examine the correlation between the expression of *RRAD* and the cell surface tension in cancer cell lines, the expression of *RRAD* gene were low in the cancer cells in our study and difficult to see the correlation with the DTD abundances (see left figure).

However, as previously reported, the regulation of cytoskeleton is highly related to the glycolysis in cancer cells, which maintain high glycolytic rates (Park, J. S., *Nature*, 578:621, 2020). *RRAD* expression is down-regulated by DNA methylation in malignant lung and breast cancers (Suzuki, M., et al., *Annals of surgical oncology*, 14:1397, 2007), suggesting that regulation of cell surface tension in *RRAD* axis may be important in some cancer cells.

We have thus added the above information in the Discussion section as follows:

“As previously reported, the regulation of cytoskeleton is also highly related to the glycolysis in cancer cells, which maintain high glycolytic rates (Park, J. S., Nature, 578:621, 2020). RRAD expression is down-regulated by DNA methylation in malignant lung and breast cancers (Suzuki, M., et al., Annals of surgical oncology, 14:1397, 2007), implying that regulation of cell surface tension by RRAD may be important in some cancer cells. We note that although we have examined the correlation between the RRAD expression and the cell surface tension with MCF-7, MDA-MB-231, and PC-3 cells, we could not confirm the correlation in our current dataset.”

9. In Fig. 1b, authors proved that Cytochalasin D-mediated loss of in-plane membrane tension in the noninvasive MCF10A cells caused a decrease of detectable signals. However, it might be because surface tension is high in MCF10A cells, more data on the influences of loss of cortical tension in low surface tension level cells need to be provided to illustrate the working detect range of cell surface tension levels. Also, the authors need to prove the influence of actin-membrane adhesion on the detectable signals, which is another essential factor influencing the cell surface tension.

We have reevaluated our manuscript regarding the influence from cortical tension and membrane tension. Although in the original manuscript we separately discussed them, it was misleading because ELASTomics is unable to differentiate the contributions from the cortical tension and cell membrane tension, which are closely related and are regulated by each other (Gauthier, N. C., et al., *Trends in cell biology*, 22:527, 2012). Instead of individually measuring cortical tension and membrane tension, ELASTomics measures the cellular mechanics abstracting as the cell surface tension, which is influenced by multiple factors including the regulation of actin-membrane adhesion proteins such as the ERM protein, the cortical tension, which is mainly the cytoskeleton, and the membrane tension of the phospholipid bilayer. We think scrutinizing the influences from all of the individual factors other than those included in the current study are subjects for future research. Although we do not include data examining the influence of the actin-membrane adhesion, we additionally performed nanopore electroporation with HeLa cells treated with blebbistatin to examine if our approach can detect the change in the cell surface tension. Blebbistatin, a myosin II-specific ATPase inhibitor, and Y-27632, a selective ROCK inhibitor, are also one of the important players in the cell surface mechanics. The blebbistatin treatment on HeLa cells decreased the quantity of imported molecules by nanopore electroporation, demonstrating again that our approach is capable to track changes in the cortical tension through cytoskeletal changes (Supplementary Fig. 5e).

We also demonstrated that methyl- β -cyclodextrin (M β CD), the drug that precisely removes cholesterol from the cell membrane, decreased the amount of molecules imported by nanopore electroporation in HeLa cells, indicating that our approach tracks changes in cell surface tension through the contribution of the phospholipid bilayer (Supplementary Fig. 5f).

As we discussed in the response #5 of Reviewer#1's comment, the sensitivity is tuneable by changing the magnitude of the applied voltage (Supplementary Fig. 3).

We have added these new experimental data in Result section as follows:

“Additionally, we confirmed that the nanopore-electroporation can probe the perturbation in the cell surface tension by blebbistatin and Y-27632, which changes the cortical tension, (Supplementary Fig. 5e) and methyl- β -cyclodextrin, which changes the plasma membrane tension (Supplementary Fig. 5f), suggesting the robustness of the approach.”

We also added the method of drug treatment in Method section as follows:

“To reduce the cortical tension by disrupting the regulation of the actin cytoskeleton, HeLa cells were incubated with 10 μ M cytochalasin D (037-17561, FUJIFILM), (-)-Blebbistatin (021-17041, FUJIFILM), or Y-27632 (10005583, CAYMAN) in PBS for 120 minutes at room temperature. To change the plasma membrane tension, cholesterol in the plasma membrane was removed by methyl- β -cyclodextrin (M β CD). 250 mM M β CD (332615, Sigma-Aldrich) in Milli-Q water was rotated with or without 50 mM cholesterol (C8667, Sigma-Aldrich) for 1 h, diluted to 10 mM with serum-free DMEM, and then filtered through 0.22 μ m Millex-GP (SLGPR33RS, Merck). Cells were incubated with DMEM containing 10 mM M β CD or M β CD/cholesterol for 1 h at 37°C.”

We also discussed the details of the detectable range of the cell surface tension in the response to the comment #10.

10. In Fig. 1c and line 167, critical sentences describing the detailed values of detectable signals are missing. Only by indicating one group is higher than the other one is inadequate to illustrate the accurate detecting efficiency of the system. Similar issues also occur in other figures like Fig. 1d. In line 109, the authors indicated that results acquired from optical tweezers are similar to the ones from proposed system. However, the comparison between the two results and advantages over the optical tweezers is missing.

We agree with the reviewer to add discussion on the detectable range. Compared to optical tweezers and other methods such as micropipette aspiration, absolute quantitation of the cell surface tension by ELASTomics is difficult, but ELASTomics is uniquely capable of simultaneously measuring the gene expression and cell surface tension of single cells at high throughput.

To quantitatively discuss the range of detectable cell surface tension, we have added the plasma membrane tensions reported in a previous study for reference:

“whose plasma membrane tensions were reported as 91.89 pN/ μ m, 82.78 pN/ μ m, 45.19 pN/ μ m (ruffling) (50.45 pN/ μ m blebbing), and 38.33 (ruffling) (42.61 pN/ μ m, blebbing) (Tsujita, K., et al., Nat Commun, 12: 5930, 2021).”

In addition, to quantitatively estimate the accuracy of ELASTomics, we performed a regression analysis on the data shown in Fig. 1d, resulting the standard error as $10^{\pm 0.39}$ pN/ μ m. We hypothesize the accuracy of the measurement of the cell surface tension is the same order of the standard error.

We have added the accuracy of measurement in a result section: *“The expected error in the measurement of the cell surface tension using the total quantity of the imported molecules was estimated on the order of $10^{\pm 0.39}$ fold in the range of 10-1000 pN/ μ m on the basis of the correlation among them.”*

11. In line 135, The title is too brief. Only listing two subjects for analysis is inadequate to conclude the following results. Same issues occur in the following titles.

We changed the section and section title to be more specific (also see our response to #4).

-In line 159, authors chose genes regarding MHC II binding and symporter activities to underscore that ELASTomics detected unique target proteins that are nonsignificant in conventional fold change of the gene expression detection. However, in Fig. S5j, similar genes regarding sodium ion transporter were neglected and not discussed, which also influence the surface tension.

We agree to add the discussion. Changing intracellular ionic concentration via ionic channels and transporters induces changes in osmotic pressure and cell surface tension (Chadwick SR, Wu JZ, Freeman SA., *Cell Physiol Biochem.*, 2:24, 2021; Xie, K., Yang, Y., & Jiang, H., *Biophysical journal*, 114:675, 2018).

We have added this information in Discussion section as follows:

*“GSEA showed enrichment not only for cytoskeleton-related gene sets such as MHC class II protein complex binding, but also for symporter activity and sodium ion transmembrane transporter activity (Supplementary Fig. 8c). Changes in intracellular ion concentrations induce changes in osmotic pressure and cell surface tension (Chadwick SR, Wu JZ, Freeman SA., *Cell Physiol Biochem.*, 2:24, 2021; Xie, K., Yang, Y., & Jiang, H., *Biophysical journal*, 114:675, 2018). Thus, ionic channels and transporters may contribute to changes in cell surface tension by transporting intracellular ions to alter osmolarity.”*

-Meanwhile, it seems that ELASTomics failed to show significant differences for essential genes that regulate the surface tension, including cadherin binding, cell adhesion molecule binding,

actin binding and so on, which are significantly different with conventional fold change detection. Therefore, the accuracy of the ELASTomics detection on potentially meaningful marker genes is questionable.

We additionally performed gene ontology (GO) enrichment analysis with the correlated genes using the MCF7 and MDA-MB-231 cells and identified enriched functions including actin binding, cadherin binding, myosin heavy chain binding, myosin binding, cell-cell adhesion mediator activity, and MHC class II protein complex binding (Supplementary Fig. 8a, b), indicating that ELASTomics can detect well-known essential genes.

We have added the GO enrichment analysis in Result section:

“Gene ontology (GO) enrichment analysis in positively (correlation >0.2) or negatively (correlation <-0.15) correlated genes showed enrichment in genes that regulate the cell adhesion, including cadherin binding and cell adhesion mediator activity (Supplementary Fig. 8a), and essential genes that regulate the cell surface mechanics, including actin binding, myosin binding. (Supplementary Fig. 8b).”

12. In line 217, authors utilized 50V for electroporation of fibroblast, while in line 64, authors used 40V for cancerous cells, and 75V for nonadherent cells. The selection standard and optimization of the electroporation voltage for the best detection outcome is missing in the text.

As we discussed in the response #5 of Reviewer#1’s comment, we added the new section about the determination of experimental condition in result section.

13. In line 177, gene names should be normatively written as SCA1, CD48, and CD150. In line 569, equation is incomplete to show. These typos should be carefully checked.

Thanks for pointing out our mistake. We have corrected Cd48 and Cd150 to CD48 and CD150. We are afraid that Sca-1 (Ly6a) is the common name for a mouse protein, so we have used Sca-1.

Reviewer #3 (Remarks to the Author):

In this manuscript the authors describe a new technique called ELASTomics that can simultaneously profile cell surface mechanics information and tie it to a cell's transcriptomic signature at single cell resolution. For this study the authors have designed DNA tagged dextran (DTD) molecules of different sizes which are then electroporated into cells using track etched membrane platforms. The intracellular DTD molecule abundance can then be read out by sequencing the tagged DNA as part of the cell's transcriptome and proteome (CITE-seq). The cargo delivery quantity via nano-electroporation directly correlates to cell surface tension and thus serves as a surrogate for cell surface tension information. Hence, this technique can be used as a method to study cell surface mechanics and connect it to gene expression.

The authors first characterize the ELASTomics method by showing the positive correlation between DTD abundance and cell membrane tension by chemically modifying cytoskeletal proteins to reduce membrane tension and comparing the results to controls. They also show the same correlation in different cell types with known differences in membrane tension. Next, they use ELASTomics to quantify the transient differences in surface mechanical properties of hematopoietic stem/progenitor cells during erythroid differentiation. They find that membrane tension increases along the erythroid lineage and genes like spectrin alpha and beta that regulate erythroid membrane structure and membrane tension are positively correlated to DTD counts. Finally, they investigate the change in cell membrane tension with increasing cellular senescence using ELASTomics. They find that cell membrane tension increases with increasing senescence and genes like RRAD contribute to increasing membrane tension and senescence.

Overall, this manuscript is well written, and the experiments are well executed. Although nano-electroporation is an existing method, its combination with a reporter for cell surface tension and transcriptomics is novel and provides an interesting method for investigating structural mechanics of cells along with their gene expression at the single cell level. The experimental designs and data analysis are adequate. The inferences and conclusions are sufficiently supported by the results. The methodology is sound and described in detail which should allow reproducibility. However, the following comments need to be addressed prior to publication.

We thank the reviewer for the positive evaluation. We were encouraged by the reviewer's enthusiasm for our approach. In our revised manuscript, we have addressed all the concerns in full by adding new experiments and analyses. We respond to the questions in a point-by-point format as follows:

- 1) A brief discussion of previous literature of nano-pore electroporation is recommended in the introduction to provide context:

- a) W. Kang et al., Microfluidic device for stem cell differentiation and localized electroporation of postmitotic neurons, *Lab on a Chip*, 2014
- b) Cao et. al., Nontoxic nanopore electroporation for effective intracellular delivery of biological macromolecules, *PNAS*, 2019
- c) Patino et. al., Multiplexed high-throughput localized electroporation workflow with deep learning–based analysis for cell engineering, *Science Advances*, 2022
- d) Pathak et. al., Cellular Delivery of Large Functional Proteins and Protein–Nucleic Acid Constructs via Localized Electroporation, *Nano Letters*, 2023
- e) Nathamgari et al., Localized electroporation with track-etched membranes, *PNAS*, 2019

Thank you for the suggestion. We added the information about the nanopore–electroporation to the introduction section as follows:

“ELASTomics utilizes nanopore electroporation, which imports molecules into cells in a manner dependent on the cell surface tension (Pathak et. al., Nano Letters, 23:3653 2023; Patino, C. A., et al., Science Advances, 8:eabn7637, 2022; Cao, Y., et al., PNAS, 116:7899, 2019; Nathamgari, S. S. P., et al., PNAS, 116:22909 2019; Kang, W., et al. Lab on a Chip 14:4486, 2014).”

2) The authors use 100 nm sized nanopores for the experiments. What pore density was used for the studies? Pore density has a significant impact on the electroporation parameters and the cargo delivery dosage. So, this value must be reported in the main text.

We have added the information about the pore density to the Method section as follows:

“(Isopore, VCTP04700, Merck; pore size: 100 nm; pore density: $6.04 \pm 0.52 \mu\text{m}^{-2}$)”

“(ipCELLCULTURE, 2000M23/620N403/13, it4ip; pore size: 400 nm; pore density: $2.0 \times 10^{-2} \mu\text{m}^{-2}$)”

3) In Fig 1h, the MCF10A cluster seems to be separated by the voltage applied (0 V control vs 40 V) with the 40 V treated cell cluster showing DTD expression as expected. However, could the authors discuss why there might be this separation of the 2 groups in UMAP clustering? In this context previously it has been shown that nano-electroporation could elevate cell stress transiently – a) Mukherjee et. al., Single cell transcriptomics reveals reduced stress response in stem cells manipulated using localized electric fields, *Materials Today Bio*, 2023.

Could there be some cell stress gene overexpression that is causing this clustering separation? The impact of electroporation induced stress should be discussed in context of the previous reports and the inferences drawn in this study.

As we discussed in the response #2 and #3 of Reviewer#1’s comment, we also affirm that a high voltage affects the gene expression.

To minimise the perturbation in the gene expression due to the nanopore electroporation, we have conducted optimisation in terms of applied voltages, frequency, and buffer conditions (Supplementary Fig. 3). Further, to focus on the difference in the gene expression due to the cell surface tension, we prepared scRNA-seq with non-nanopore electroporated cells for reference and integrated them with nanopore electroporated cells before the analysis. As the reviewer pointed out, in Fig.1h, the MCF10A cluster of nanopore electroporated cells seems to be slightly separated from the non-nanopore electroporated ones. As recommended, we have additionally performed an analysis comparing between electroporated and non-electroporated MCF10A cells (Supplementary Fig. 8d). Consistent with the previous work (Mukherjee, P., et al., *Materials Today Bio*, 19:100601, 2023), GSEA in MCF10A cells showed enrichment in similar functions, such as the cellular response to stress, cell death, and regulation of cell death (Supplementary Fig. 8d), indicating that 40 V was intense to MCF10A cells.

We added them in Result section as follows:

“We note that gene expression in MCF10A cells were slightly perturbed by the nanopore-electroporation (Fig. 1g-i). GSEA showed enrichment in the cellular response to stress, cell death, and regulation of cell death (Supplementary Fig. 8d), consistent with a previous work (Mukherjee, P., et al., Materials Today Bio, 19:100601, 2023).”

4) In Fig 1j the authors correlated the DTD counts to the gene expression ratio in two cell types. Could the authors identify some key genes that show statistically significant positive correlation and discuss the biological implications?

As we discussed in the response #11 of Reviewer#2’s comment, we additionally performed gene ontology (GO) with the positively correlated genes (Supplementary Fig. 8b).

We have added them in discussion section as follows:

As we discussed in the response #11 of Reviewer#2’s comment, we added the new section about GO enrichment analysis with the positively correlated genes in result section.

-Similarly, how they have done for subsequent erythroid differentiation and cellular senescence studies.

We have discussed the data analysis on erythroid differentiation and cellular senescence in the response to #5.

5) For the HSPC differentiation and cellular senescence studies a pathway analysis like GSEA is recommended like was done for the cell line comparison. It may reveal additional insights into the biological processes like in the first study.

We appreciate the very interesting insight by the reviewer. We have performed the additional GSEA on the data of the mHSPCs differentiation and cellular senescence studies.

GSEA of mHSPCs relating to erythroid differentiation showed enrichment in the functions related to spectrin binding (GO:0030507), actin binding (GO:0003779), ATP-dependent activity (GO:0140657), and ATP hydrolysis activity (GO:0016887) (Supplementary Fig. 10). Given that spectrin is regulated by glycolysis via ATP-dependent phospholipid flippase as well as membrane-cytoskeletal linkers (Manno, S., et al., *Journal of Biological Chemistry*, 285:33923, 2010), the GSEA result imply that the transient increase in cell surface tension involved the ATP-dependent cytoskeletal regulation.

We added them in discussion section as follows:

“GSEA also showed enrichment in spectrin binding (GO:0030507) and gene sets which related to the regulation of spectrin via ATP-dependent phospholipid flippase (Manno, S., et al., Journal of Biological Chemistry, 285:33923, 2010). (Supplementary Fig. 10).”

GSEA of TIG-1 cells showed enrichment not only aging (GO:0007568) but also actin binding (GO:0003779), ATP-dependent activity (GO:0140657), and ATP hydrolysis activity (GO:0016887) (Supplementary Fig. 12g-k). The result support our hypothesis in the Discussion that the correlation between *RRAD* expression and increased cell surface tension by the decrease in ATP and blunted regulation of cytoskeleton-related proteins by inhibition of the glycolysis pathway.

We have added these results in the discussion section as follows:

“GSEA also showed enrichment in glycolysis-related gene sets such as ATP-dependent activity (GO:0140657), and ATP hydrolysis activity (GO:0016887) as well as aging (GO:0007568) (Supplementary Fig. 12g-k).”

REVIEWER COMMENTS

Reviewer #1 (Remarks on code availability):

The authors have addressed all my concerns. However, although the authors shared the code on GitHub, it is very difficult to know how to use it. The authors should provide more detailed documentation and tutorials.

Reviewer #2 (Remarks to the Author):

The authors have improved their manuscript with new experiments and results. Yet, the following questions remain not well addressed.

1. Whether the notable cell-to-cell heterogeneity in DTD counts within and across different cell types measured by ELASTomics is caused by the fluctuation of the measurement remains unclear.
2. The general role of RRAD in regulating cell surface tension and functional changes needs to be defined. The UAMP figure in the rebuttal letter is hard to understand without explanation.
3. The authors didn't address well why they chose to use TIG-1 cell line rather than primary fibroblast, Primary Lung Fibroblasts. Simply listing the limitation didn't address the issue. Meanwhile, the reference cited in rebuttal used to explain the differences in two cell types discusses rodent fibroblasts rather than human fibroblasts. Primary human fibroblast is easily accessible (COPD PCS-201-017™, ATCC). So, the justification why primary cells were not used is weak and may need further data support.

Reviewer #3 (Remarks to the Author):

The reviewers addressed all my comments. The manuscript is ready for publication.

Reviewer #4 (Remarks to the Author):

The authors have adequately addressed all the comments. The recommendation is to publish the

manuscript in Nature Communications in its current format.

Reviewer #1 (Remarks on code availability):

The authors have addressed all my concerns. However, although the authors shared the code on GitHub, it is very difficult to know how to use it. The authors should provide more detailed documentation and tutorials.

We thank the reviewer for pointing out this. We have organised the code in GitHub (in the branch of “Shiomi.et.al2023” at <https://github.com/RIKEN-Microfluidics-Lab/ELASTomics.git>) and added a readme file to use them. We also added a demo program and data for the tutorial.

Reviewer #2 (Remarks to the Author):

The authors have improved their manuscript with new experiments and results. Yet, the following questions remain not well addressed.

We thank the reviewer for the evaluation. We have further re-evaluated our manuscript and have tried to address all the points raised by the reviewer in full. We have responded to the comments in a point-by-point format as follows:

1. Whether the notable cell-to-cell heterogeneity in DTD counts within and across different cell types measured by ELASTomics is caused by the fluctuation of the measurement remains unclear.

As discussed in the previous revision, we showed that the amount of the molecules delivered into cells via nanopore electroporation consistently correlated with the cell surface tension (Fig. 1d and Supplementary Fig. 4h, i). We also added experimental results using atomic force microscopy with TIG-1 cells of different senescence levels (Fig. 3b and Supplementary Fig. 11d).

We understand that the reviewer is concerned about the heterogeneity in DTD counts, which we did not explicitly discuss in the previous revision. In ELASTomics, we used five types of DTDs with different molecular weights. We observed highly consistent counts despite the quantification of DTDs being independent, supporting the reliability of the DTD counts (Supplementary Fig. 6). In our revised manuscript, we have explicitly explained the consistency and reliability of DTD counts as:

Line 151:

“Notably, the permeability of DTDs with different molecular weights was highly consistent (Supplementary Fig. 6), supporting the reliability of the DTD counts for the quantification regardless of the cell types.”

2. The general role of RRAD in regulating cell surface tension and functional changes needs to be defined. The UAMP figure in the rebuttal letter is hard to understand without explanation.

Thank you for your suggestion. In this work, we focused on reporting the ELASTomics and showcasing the applications to various types of cells in various contexts (cancer cells (Fig.1), primary hematopoietic stem/progenitor cells (Fig.2), cellular senescence (Fig.3), cells from various species (Supplementary Fig. 3 p-r)) rather than exploring the general role of the *RRAD* in the regulation of cell surface tension.

However, we agree with the reviewer that exploring the general role of the *RRAD* in the regulation of cell surface tension is important and further work needs to be done in order to ask if the finding with TIG-1 cells is generalisable or cellular senescence-specific. In this work, we have shown that *RRAD*, a gene that suppresses glycolysis by inhibiting the translocation of GLUT1 to the plasma membrane, is involved in increasing cell surface tension in TIG-1 cells along cellular senescence. Interestingly, the promotion of GLUT1-mediated glycolysis via inhibition of *TXNIP* also changes the cell mechanics in some cancer cells (Sullivan, W. J. et al., *Cell*, 175:117, 2018). Indeed, our data showed a positive correlation between cell surface tension and *TXNIP* expression in cancer cell lines. However, our approach failed to detect the expression of *RRAD* in cancer cells owing to the low sensitivity of scRNA-seq with Chromium (Supplementary Fig. 13). Glycolysis is intimately involved in regulating the cytoskeleton, which consumes large amounts of energy. We thus envision that inhibition of the glycolysis pathway, including upregulation of *RRAD*, may be generally involved in the regulation of cell surface tension in various contexts.

Thus, we have added our vision on the exploration of the general role of *RRAD* in the discussion as:

“It has also been reported that TXNIP, which inhibits translocation of GLUT1 to the plasma membrane like RRAD, alters cell mechanics in some cancer cells (Sullivan, W. J., et al., Cell, 175:117, 2018). Indeed, we found a positive correlation between cell surface tension and TXNIP expression in cancer cell lines (Supplementary Fig. 13a, b). We thus envision that inhibition of glycolysis pathway, including upregulation of RRAD, may play a general role in the regulation of cell surface tension in various contexts. We note that although we have examined the correlation between the RRAD expression and the cell surface tension with MCF7, MDA-MB-231, and PC-3 cells, we could not confirm the correlation in our current dataset owing to the low sensitivity of scRNA-seq (Supplementary Fig. 13c).”

We apologise for our poor explanation on the UMAP figure in the previous point-by-point response. We re-evaluated the UMAP figure and have added a new Supplementary Figure 13c, which shows poor detection of *RRAD* gene in our four cancer cell lines (MCF10A: 12.8%; MCF7: 0.0%; MDA-MB-231: 15.3%; PC-3: 0.0%).

3. The authors didn't address well why they chose to use TIG-1 cell line rather than primary fibroblast, Primary Lung Fibroblasts. Simply listing the limitation didn't address the issue. Meanwhile, the reference cited in rebuttal used to explain the differences in two cell types discusses rodent fibroblasts rather than human fibroblasts. Primary human fibroblast is easily accessible (COPD PCS-201-017™, ATCC). So, the justification why primary cells were not used is weak and may need further data support.

We agree with the reviewer that primary fibroblasts isolated directly from humans may retain many functional features in ageing. Comparative analysis between patient primary cells, such as primary lung fibroblasts isolated from COPD (chronic obstructive pulmonary disease) patients and primary lung fibroblasts isolated from healthy individuals, would provide insights directly relevant to the ageing-related disease.

In contrast, TIG-1 cell is a human diploid fibroblast-like cell strain with a normal female karyotype. It was isolated from a human fetal lung for the study of cellular senescence (Ohashi, M., et al., *Exp. Gerontol.*, 15:121, 1980). A differential proteomic analysis of TIG-1 cells identified *ATP6V0A2*, the causative gene for autosomal recessive cutis laxa type 2 (ARCL2), which has been implicated as a responsible gene for abnormal glycosylation and Golgi transport defects along cellular senescence (Udono, M., et al., *Cell Rep.*, 5:17342, 2015). TIG-1 cells has been also utilised to study abnormal lipid accumulation in hepatocytes due to decreased expression of *SMARCD1* along cellular senescence (Inoue, C., et al., *NPJ Aging Mech. Dis.*, 3:11, 2017), and ageing-related mitochondrial dysfunction and excessive mitochondrial reactive oxygen species production (Fujita Y., et al., *Exp Gerontol.*, 165:111866, 2022). Accordingly, owing to the low genetic variation and high reproducibility, TIG-1 cell is advantageous when screening specific genes involved in cell surface tension along cellular senescence.

Regarding the applicability of ELASTomics to primary cells, we have demonstrated the application to hematopoietic stem/progenitor cells isolated from mouse bone marrow (Fig. 2).

To discuss why we chose to use TIG-1 cells rather than primary fibroblast, we have elaborated information on TIG-1 cells in the result section and discussion section as follows:

“We performed ELASTomics with human foetal lung fibroblasts (TIG-1), which exhibit replicative senescence accompanied by changes in cholesterol abundance in lipid rafts, chromosomal instability, abnormal glycation and Golgi transport, abnormal lipid accumulation, and mitochondrial dysfunction as cellular senescence progresses.”

“TIG-1 cell was isolated from a human fetal lung for the study of cellular senescence. TIG-1 cells have also been utilised to study ageing-related phenomena, including mitochondrial dysfunction, excessive reactive oxygen species production, and abnormal lipid accumulation. An analysis utilised TIG-1 cell identified that ATP6V0A2, a gene responsible for autosomal recessive cutis laxa type 2 (ARCL2), triggers abnormal glycosylation and Golgi transport as a cellular senescence program. Accordingly, owing to the low genetic variation and high reproducibility, TIG-1 cell is advantageous when screening specific genes involved in cell surface tension along cellular senescence.”

Reviewer #3 (Remarks to the Author):

The reviewers addressed all my comments. The manuscript is ready for publication.

Thank you for taking the time to review our manuscript.

Reviewer #4 (Remarks to the Author):

The authors have adequately addressed all the comments. The recommendation is to publish the manuscript in Nature Communications in its current format.

Thank you for taking the time to review our manuscript.

REVIEWERS' COMMENTS

Reviewer #1 (Remarks to the Author):

The authors have addressed all my concerns, and I would recommend the paper be accepted for publication.

Reviewer #2 (Remarks to the Author):

The authors have adequately addressed all the comments.